# ADAMTS4 is a crucial proteolytic enzyme for versican cleavage in the amnion at parturition
Meng-Die Li[1,2], Jiang-Wen Lu[1,2], Fan Zhang[1,2], Wen-Jia Lei[1,2], Fan Pan[1,2], Yi-Kai Lin[1,2], Li-Jun Ling[3], Leslie Myatt[4], Wang-Sheng Wang [1,2] ✉ & Kang Sun [1,2] ✉

Hyalectan cleavage may play an important role in extracellular matrix remodeling. However, the proteolytic enzyme responsible for hyalectan degradation for fetal membrane rupture at parturition remains unknown. Here, we reveal that versican (VCAN) is the major hyalectan in the amnion, where its cleavage increases at parturition with spontaneous rupture of membrane. We further reveal that ADAMTS4 is a crucial proteolytic enzyme for VCAN cleavage in the amnion. Inflammatory factors may enhance VCAN cleavage by inducing ADAMTS4 expression and inhibiting ADAMTS4 endocytosis in amnion fibroblasts. In turn, versikine, the VCAN cleavage product, induces inflammatory factors in amnion fibroblasts, thereby forming a feedforward loop between inflammation and VCAN degradation. Mouse studies show that intra-amniotic injection of ADAMTS4 induces preterm birth along with increased VCAN degradation and proinflammatory factors abundance in the fetal membranes. Conclusively, there is enhanced VCAN cleavage by ADAMTS4 in the amnion at parturition, which can be reenforced by inflammation.

Despite advances in perinatal care, preterm birth continues to be the leading cause of mortality in children under age 5[1]. Understanding the mechanism underlying preterm birth is of paramount importance for the development of effective strategies for the prevention of preterm birth. It is acknowledged that preterm prelabor rupture of the membranes (PPROM) accounts for one percent of all pregnancies, but for thirty to forty percent of preterm birth[2]. Moreover, PPROM is the leading identifiable cause of preterm birth. However, the mechanism underlying PPROM remains poorly understood. Elucidation of the mechanism may help find ways to prevent PPROM thus reducing the rate of preterm birth.

The human fetal membranes are a viscoelastic structure comprised of two morphologically distinct layers, the inner amnion and the outer chorion[2-4]. Although the outer layer, which makes extensive contact with the uterine wall, may develop from differential germ layers of the embryo in different species, there is always an inner amnion layer surrounding the embryo regardless of species[5]. It is the amnion layer where the tensile strength of the fetal membranes is mostly derived from[6]. This feature of the amnion explains why the amnion usually ruptures last during membrane rupture at parturition. Thus, understanding how the amnion undergoes extracellular

matrix (ECM) remodeling in membrane rupture is undoubtedly of clinical significance for the prevention of PPROM. The amnion is further comprised of the epithelium, basement membrane, compact layer of connective tissue, fibroblast layer and intermediate spongy layer from innermost to outermost[2-4]. The ECM of the amnion is composed of an interlocking mesh of fibrous proteins (mostly collagen) as well as non-fibrous proteoglycans[7-9]. Although the collagen content of the ECM has been investigated extensively, few studies have addressed the role of the non-fibrous proteoglycans in membrane rupture. Proteoglycans are important non-fibrous components of the ECM, which not only stabilize ECM structure by attracting water molecules to moisten ECM and resident cells, but also trap growth factors[10-12]. Proteoglycans are further classified into heparan sulfate, dermatan sulfate, chondroitin sulfate, keratan sulfate proteoglycans and hyaluronic acids based on the size and nature of their glycosaminoglycan chains[13,14]. Among those, the hyalectan class of chondroitin sulfate and keratan sulfate proteoglycans is emerging as an important mediator of a variety of embryogenic activities[15-19]. The hyalectan family of chondroitin sulfate and keratan sulfate proteoglycans is comprised of aggrecan (ACAN), neurocan (NCAN), brevican (BCAN) and versican (VCAN). The stabilizing action of the hyalectan family on ECM integrity is

[1]Center for Reproductive Medicine, Ren Ji Hospital, School of Medicine, Shanghai Jiao Tong University, Shanghai, PR China. [2]Shanghai Key Laboratory for Assisted Reproduction and Reproductive Genetics, Shanghai, PR China. [3]Department of Obstetrics, Shanghai First Maternity and Infant Hospital, School of Medicine, Tongji University, Shanghai, PR China. [4]Department of Obstetrics and Gynecology, Oregon Health & Science University, Portland, OR, USA. ✉e-mail: wangsheng_wang@hotmail.com; sungangrenji@sjtu.edu.cn

lost upon proteolytic cleavage[20]. However, the proteolytic enzyme responsible for their cleavage in membrane rupture remains unknown. Although MMPs (matrix metalloproteinases)[21], plasmin[22] and ADAMTS (a disintegrin and metalloprotease domains with thrombospondins motif)[23,24] are all reported to possess proteolytic activities toward the hyalectan members, the multi-domain proteinase ADAMTS is believed to play a central role in the turnover of the hyalectan members. Of the 19 members of the ADAMTS family in humans, 7 members (ADAMTS1, 4, 5, 8, 9, 15, and 20) have been identified to carry extracellular proteoglycanase activity toward the hyalectan members[25,26]. However, it remains to be determined whether these ADAMTS isoforms are involved in the cleavage of the hyalectan family for membrane rupture at parturition. Notably, ADAMTS abundance in the interstitial tissue is determined not only by ADAMTS expression, but also by its endocytosis via the endocytic cell surface receptor, i.e., the low-density lipoprotein receptor-related protein 1 (LRP1)[27]. LRP1 shedding from the cell surface will increase ADAMTS concentrations in the interstitial tissue by reducing its endocytosis thus enhancing ADAMTS accumulation in the interstitial tissue[28,29].

Inflammation of the fetal membranes is an indispensable event of parturition[30,31]. Based on the presence or absence of infection, inflammation of the fetal membranes can either be infectious or non-infectious in nature, also known as infectious and sterile chorioamnionitis respectively[32–34]. While infectious chorioamnionitis is more common in infection-induced preterm birth, sterile chorioamnionitis is usually encountered in normal parturition at term, as well as in some spontaneous preterm birth[35]. Inflammatory factors produced in chorioamnionitis potently induce the synthesis of protease and prostaglandins, the mediators of ECM remodeling in membrane rupture, cervical ripening and myometrial contraction at parturition[36–39], despite that inflammation is also known as an inducer of tissue fibrosis in other parts of the body[40]. However, it is not clear whether inflammatory factors produced in chorioamnionitis facilitate VCAN cleavage through induction of ADAMTS accumulation in the interstitial tissue. Of interest, ECM cleavage products can act as a damage-associated molecular pattern molecule to evoke inflammatory reactions in both gestational and non-gestational tissues[41–43]. Hence, it is tempting to postulate that the hyalectan cleavage product may also be involved in the inflammatory reactions of the human amnion at parturition.

Based on the rationale described above, we hypothesized that proinflammatory factors produced in chorioamnionitis might contribute to ADAMTS accumulation in the interstitial tissue of the fetal membranes via induction of ADAMTS expression and inhibition of ADAMTS endocytosis at parturition, thus leading to enhanced hyalectan degradation. The hyalectan cleavage products may further enhance the inflammatory reactions of the membranes, thus setting up a feedforward cycle between inflammation reactions and hyalectan cleavage in the fetal membranes, which ultimately weakens the membrane for rupture at parturition. Here, we addressed this hypothesis by using human amnion tissue, the most tensile layer of the membranes, obtained from deliveries with spontaneous rupture of membranes (SROM) and in cultured human amnion tissue explants as well as in primary human amnion fibroblasts, the major source of ECM components. Finally, a mouse model was developed to investigate whether intra-amniotic injection of ADAMTS could increase hyalectan cleavage in the fetal membrane at parturition in vivo.

## Results
### Reciprocal changes in VCAN and versikine in the human amnion in deliveries with SROM

Analysis of our previously published transcriptomic sequencing data (NCBI GEO accession number GSE166453)[44] revealed that *VCAN* was abundantly expressed while the transcripts of the other members (*ACAN, NCAN, BCAN*) of the hyalectans family were hardly detectable in human amnion (Fig. 1a, Supplementary Fig. S1a). These results were subsequently confirmed with quantitative real-time polymerase chain reaction (qRT-PCR) (Supplementary Fig. S1b). Hence, VCAN was focused on in subsequent studies. Measurement with qRT-PCR showed that the abundance of *VCAN* mRNA in the human amnion was comparable between TNL (elective

cesarean section without labor, designated as term non-labor) and TL (vaginal delivery with SROM at term, designated as term with labor) groups (Fig. 1b). However, Western blotting analysis showed that the abundance of VCAN protein was significantly decreased, while the abundance of versikine protein, the cleavage product of VCAN, was significantly increased in the human amnion of the TL group as compared to the TNL group (Fig. 1c). The reciprocal alteration in VCAN and versikine protein abundance in the human amnion in TL was confirmed with immunohistochemical staining (Fig. 1d). These data indicate that the human amnion expresses mainly VCAN of the hyalectan family, which undergoes increased proteolytic cleavage in deliveries with SROM.

### Alteration of ADAMTS4 and LRP1 in the human amnion in deliveries with SROM

ADAMTSs are known to play a central role in VCAN proteolysis. Among members (ADAMTS1, 4, 5, 9, 15, and 20) of the ADAMTS family carrying proteoglycanase activity toward VCAN[25,26], ADAMTS4 was the only member manifesting a significant increase in the human amnion in TL as revealed by analysis of the transcriptomic sequencing data published previously[44] (Fig. 2a and Supplementary Fig. S2a). This increased ADAMTS4 expression in the human amnion of the TL group was subsequently confirmed at both mRNA and protein levels with qRT-PCR and Western blotting (Fig. 2b, c). Moreover, the abundance of the endocytic cell surface receptor LRP1 was significantly decreased at both mRNA and protein levels in the human amnion of the TL group as compared to that of the TNL group as revealed by transcriptomic sequencing, qRT-PCR and Western blotting (Fig. 2c, d and Supplementary Fig. S2b). These data suggest that increased ADAMTS4 abundance may be responsible for versican cleavage in the amnion at parturition with SROM.

### Distribution of ADAMTS4 and LRP1 in the human amnion

In order to study the effect and regulation of ADAMTS4 in amnion cells in vitro, we examined the distribution of ADAMTS4 and LRP1 in the amnion. Analysis of our single-cell sequencing data published previously[45] showed that ADAMTS4 was exclusively expressed in mesenchymal fibroblasts of the amnion, while LRP1 appeared to be expressed in both amnion mesenchymal fibroblasts and epithelial cells (Fig. 2e). These expression profiles of ADAMTS4 and LRP1 were confirmed with qRT-PCR in cultured human amnion fibroblast and epithelial cells (Fig. 2f). Consistently, immunohistochemical staining of the human amnion also showed that ADAMTS4 was found mainly in mesenchymal fibroblasts but scarcely in epithelial cells, while LRP1 was observed in both fibroblasts and epithelial cells (Fig. 2g). The latter was confirmed in cultured amnion epithelial and fibroblast cells with Western blotting (Supplementary Fig. S3). These data indicate that ADAMTS4 is derived mainly from mesenchymal fibroblasts of the human amnion. Given that mesenchymal fibroblasts are the well-recognized primary source of ECM molecules[4], we used primary human amnion fibroblasts to examine the effect and regulation of ADAMTS4 in subsequent studies.

### Role of ADAMTS4 in VCAN degradation and ECM remodeling in the human amnion

VCAN is able to interact with several extracellular matrix and cell surface molecules. As shown in Fig. 3a, VCAN is present on the cell surface either directly via binding to cell surface receptors or indirectly via attachment to hyaluronic acids[46]. Hence, in cultured cells, cleavage of VCAN will lead to decreased VCAN abundance in the cells and increased versikine abundance in the culture medium. Treatment of cultured human amnion fibroblasts with recombinant human ADAMTS4 (rhADAMTS4) (50 ng/mL, 24 h) significantly reduced VCAN abundance in the cells along with increased versikine abundance in the culture medium (Fig. 3b). Consistently, Western blotting and immunohistochemical staining also showed that rhADAMTS4 (50 ng/mL, 24 h) treatment of cultured human amnion tissue explants significantly decreased VCAN while increased versikine abundance in the amnion (Fig. 3c−e). Moreover, examination of the explants treated with and

without rhADAMTS4 (50 ng/mL, 24 h) with transmission electron microscopes (TEM) revealed that collagen fibrils in the ECM became distorted by ADAMTS4 treatment (Fig. 3f). However, Masson trichrome and Sirius red staining showed that the abundance of total collagen did not appear to be changed by ADAMTS4 treatment (Fig. 3g, h). These data suggest that ADAMTS4 may cause ECM remodeling in the amnion through aforementioned means rather than collagen degradation, and ADAMTS4-mediated VCAN cleavage may be a contributing factor.

### Regulation of ADAMTS4 expression and endocytosis by IL-1β and SAA1 in human amnion fibroblasts

We next examined whether inflammation of the fetal membranes may regulate ADAMTS4 expression and endocytosis in the human amnion at parturition. Interleukin-1β (IL-1β) and serum amyloid A1 (SAA1), an acute phase protein which can be synthesized locally in the fetal membranes, have been shown to play important roles in the inflammatory reactions of the fetal membranes at parturition[37,47,48]. We found that both IL-1β (0, 0.1, 1 and 10 ng/mL, 24 h) and SAA1 (0, 10, 50 and 100 ng/mL, 24 h) were capable of inducing ADAMTS4 mRNA and protein expression in a concentration-dependent manner in human amnion fibroblasts (Fig. 4a−d). However, the same concentrations of IL-1β and SAA1 failed to alter *LRP1* mRNA expression in amnion fibroblasts (Supplementary Fig. S4a, b). Instead, IL-1β (10 ng/mL, 24 h) and SAA1 (100 ng/mL, 24 h) treatment increased LRP1 shedding from amnion fibroblasts as illustrated by increased extracellular LRP1 abundance in the culture medium, and correspondingly decreased LRP1 abundance in the cells (Fig. 4e, f).

The role of LRP1 in ADAMTS4 endocytosis in human amnion fibroblasts was illustrated by increased ADAMTS4 abundance in the culture medium and decreased VCAN abundance in amnion fibroblasts with small interference RNA (siRNA)-mediated knockdown of *LRP1* expression (Fig. 5a), as well as by dual immunofluorescence staining of ADAMTS4 and early endosome antigen 1 (EEA1), a marker for early endosomes in endocytosis[49], which showed that the colocalization of ADAMTS4 and EEA1 was diminished in the cytoplasm upon siRNA-mediated knockdown of *LRP1* (Fig. 5b). As illustrated in Fig. 5c, IL-1β and SAA1 could induce ADAMTS4 expression and inhibit ADAMTS4 endocytosis by enhancing LRP1 shedding. Hence, we anticipated that these proinflammatory factors might increase ADAMTS4 accumulation in the extracellular culture medium of amnion fibroblasts resulting in increased VCAN degradation. This was supported by the results showing that IL-1β (10 ng/mL, 24 h) and SAA1 (100 ng/mL, 24 h) treatment of amnion fibroblasts led to significant accumulation of ADAMTS4 and versikine in the culture medium along with decreased VCAN abundance in the cells (Fig. 5d. e). The role of ADAMTS4 in VCAN cleavage by the proinflammatory factor was further validated by the findings that the specific ADAMTS4 antagonist[50,51] (10 μM, 24 h) significantly attenuated the induction of VCAN cleavage by IL-1β (10 ng/mL, 24 h) (Supplementary Fig. S5). These data suggest that proinflammatory factors produced in chorioamnionitis could promote VCAN degradation by increasing ADAMTS4 accumulation in the interstitial tissue via induction of ADAMTS4 expression and inhibition of ADAMTS4 endocytosis in human amnion fibroblasts.

### Stimulation of proinflammatory factors by versikine in human amnion fibroblasts

The VCAN cleavage product versikine has been reported to function as a bioactive damage-associated molecular patterns involved in

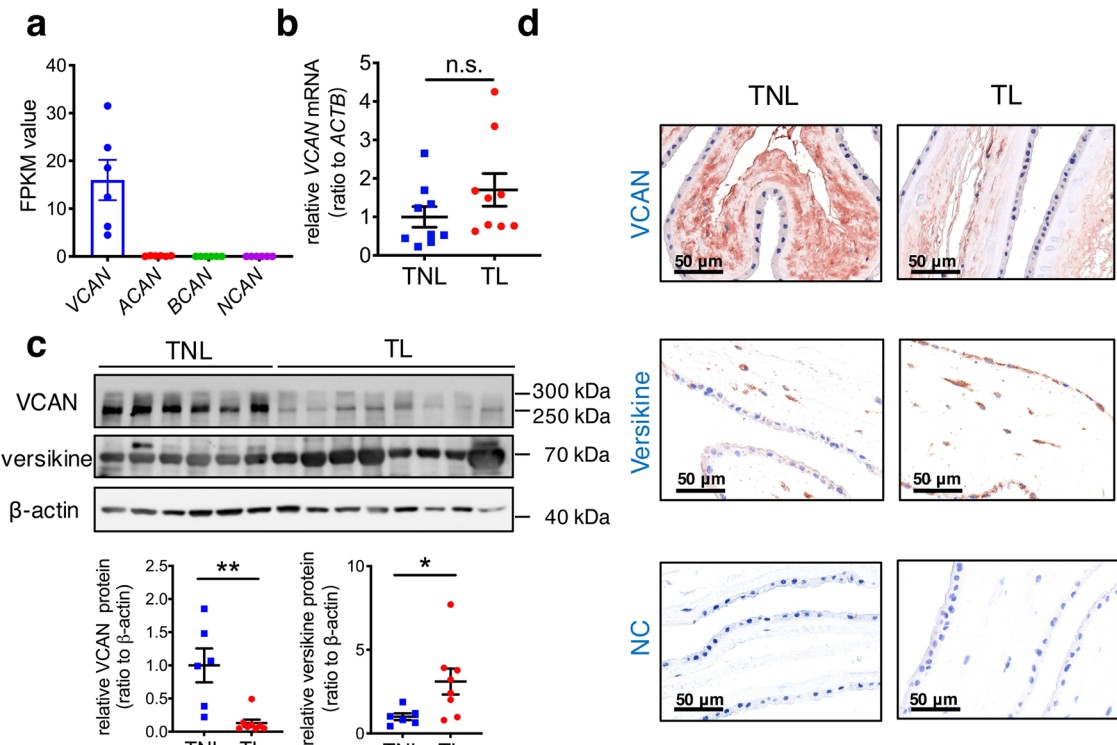

**Fig. 1 | Reciprocal changes of VCAN and versikine in the human amnion in deliveries with spontaneous rupture of membranes. a** Comparison of *VCAN*, *ACAN*, *NCAN* and *BCAN* gene transcripts in the human amnion tissues collected from term pregnancies with or without labor (*n* = 3 each). FPKM, Fragments Per Kilobase of transcript per Million mapped reads. **b** qRT-PCR analysis showing unaltered *VCAN* mRNA abundance in the human amnion collected from term labor (TL; *n* = 9) as compared to term non-labor (TNL; *n* = 9). n.s., no significance.

**c** Western blotting analysis showing reciprocal changes in VCAN and versikine protein abundance in the human amnion in TL (*n* = 8) as compared to TNL (*n* = 6). **d** Representative immunohistochemical images showing reciprocal changes of VCAN and versikine protein abundance in the amnion in TL (*n* = 3) as compared to TNL (*n* = 3). Scale bars, 50 μm. Data are mean ± SEM. Statistical analysis was performed with the Mann–Whitney U test. *$p < 0.05$, **$p < 0.01$ vs. TNL.

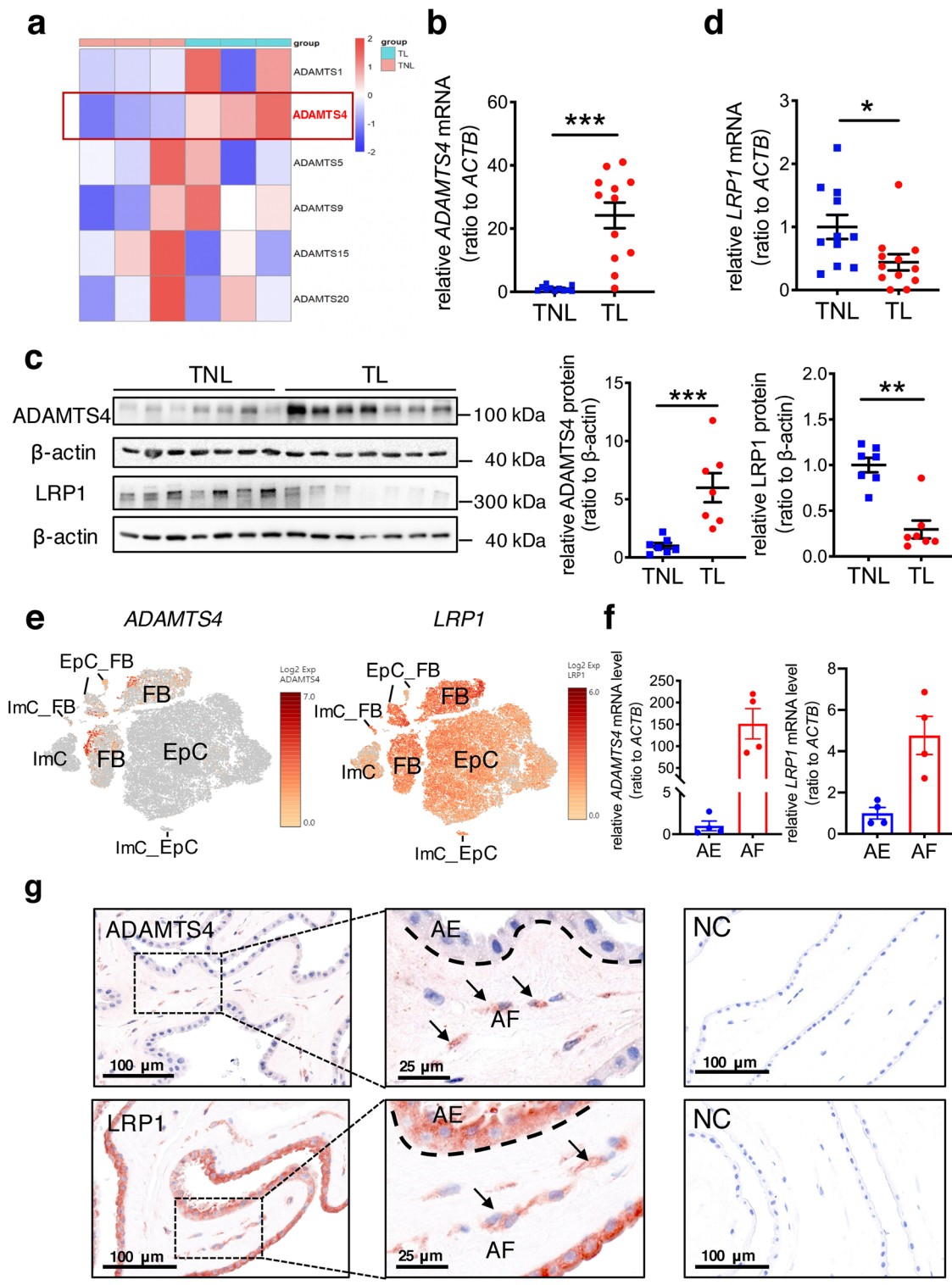

inflammatory responses in multiple non-gestational tissues[52–55]. We found that overexpression of versikine in amnion fibroblasts (Fig. 6a) increased the expression of a multitude of pro-inflammatory factors including cyclooxygenase-2 (COX-2), the rate-limiting enzyme in prostaglandin synthesis (Fig. 6b), pro-inflammatory cytokines (*IL1B, IL6, TNF*) and chemokines (*CCL2, CCL3, CCL20*) (Fig. 6c), suggesting that versikine released from VCAN cleavage may function as a damage-associated molecular pattern molecule to evoke inflammatory reactions to further enhance VCAN cleavage in the fetal membranes.

**Effects of intra-amniotic injection of ADAMTS4 on VCAN cleavage in the fetal membranes and parturition in the mouse**

The mouse fetal membranes are comprised of amnion and yolk sac membrane[56]. Immunohistochemical staining of the mouse fetal membranes showed that ADAMTS4 was present in both amnion and yolk sac layers (Fig. 7a). Western blotting measurement showed that ADAMTS4 and versikine abundance increased, and VCAN abundance decreased in the mouse fetal membranes from 14.5 days post-coitus (dpc) to 18.5 dpc (Fig. 7b) in a gestational age-dependent manner. Consistently, immunofluorescence

**Fig. 2 | Alterations of ADAMTS4 and LRP1 abundance in the human amnion in deliveries with spontaneous rupture of membranes. a** Heatmap of transcriptomic sequencing data displaying alterations in mRNA abundance of ADAMTS family members (ADAMTS1, 4, 5, 9, 15, and 20) possessing proteoglycanase activity toward VCAN in the human amnion in TL ($n = 3$) as compared to TNL ($n = 3$). Of these proteoglycanases, only ADAMTS4 manifest significantly increased expression in TL. Blue to red represents expression levels from low to high. **b** qRT-PCR analysis showing increased *ADAMTS4* mRNA abundance in the human amnion in TL ($n = 12$) compared to TNL ($n = 11$). **c** Western blotting analysis showing increased ADAMTS4 and decreased LRP1 protein abundance in the human amnion in TL ($n = 7$) compared to TNL ($n = 7$). **d** qRT-PCR analysis showing decreased *LRP1* mRNA abundance in the human amnion in TL ($n = 12$) as compared to TNL ($n = 11$). **e** Single-cell sequencing analysis displaying the cell types expressing *ADAMTS4* and *LRP1* mRNA in the human amnion. FB, fibroblasts; EpC, epithelial cells; ImC, immunocytes; EpC_FB, epithelial_fibroblasts; ImC_EpC, immune_e-pithelial cells; ImC_FB, immune_fibroblasts. **f** Comparison of *ADAMTS4* or *LRP1* mRNA abundance in cultured human amnion epithelial cells and fibroblasts prepared from TNL ($n = 4$) as measured with qRT-PCR. AE, epithelial cells; AF, amnion fibroblasts. **g** Representative immunohistochemical images showing the distribution of ADAMTS4 and LRP1 in the human amnion collected from TNL ($n = 3$). ADAMTS4 was found mainly in amnion fibroblasts, while LRP1 was found in both amnion fibroblasts and epithelial cells. Scale bars, 25 or 100 μm. Data are mean ± SEM. Statistical analysis was performed with unpaired Student's *t* test (**b**), Mann–Whitney U test (**c. d**) or paired Student's *t* test (**f**). *$p < 0.05$, **$p < 0.01$, ***$p < 0.001$ vs TNL or AE.

staining showed that weaker VCAN but stronger versikine staining were observed in the mouse fetal membranes on 18.5 dpc as compared to 16.5 dpc (Fig. 7c). Notably, intra-amniotic injection of lentiviral vector expressing *Adamts4* ($1 \times 10^6$ TU/10 μL/per gestational sac) at 16 dpc led to a 77.8% preterm birth rate (Fig. 8a−c), along with increased ADAMTS4, versikine and decreased VCAN abundance in the mouse fetal membranes (Fig. 8d). Furthermore, the abundance of *Ptgs2* (encoding COX-2)*, Il1b, Il6, Ccl2, Ccl3* and *Ccl20* mRNA was also significantly increased in the mouse fetal membranes following intra-amniotic administration of lentiviral vector expressing *Adamts4* (Fig. 8e). Observation with TEM showed that the organization of collagen fibrils in the ECM of the mouse fetal membranes was distorted by intra-amniotic administration of lentiviral vector expressing *Adamts4* (Fig. 8f).

## Discussion

The amnion layer provides the most tensile strength for the fetal membranes, which is determined mainly by mesenchymal cell-derived ECM contents[6]. Compared with the fibrous components, few studies have addressed the role of the non-fibrous proteoglycans in membrane rupture. The degradation of fibrous protein by MMPs has been investigated intensively, and was believed to be the primary driving force of membrane rupture[41,57]. However, our present study provides the evidence that ADAMTS4, a crucial proteolytic enzyme for hyalectan cleavage, the non-fibrous component of the ECM, also plays a crucial role in parturition, and ADAMTS4-mediated cleavage of VCAN in the amnion may contribute to this effect of ADAMTS4.

The hyalectan family is comprised of VCAN, ACAN, BCAN and NCAN[13]. Unlike the dominant presence of ACAN in the cartilaginous tissue[58], and NCAN and BCAN in the matrix of the central nervous system[59,60], we found VCAN rather than other family members was the dominant hyalectan in the amnion, which appears to be consistent with the wide distribution of VCAN in the developing embryo[19]. VCAN is present in multiple isoforms, which share a homologous globular N-, C-terminal, and a central glycosaminoglycan (GAG) binding domain, but vary in size and the number of GAG chains. The N- and C-terminals of VCAN maintain the integrity of the ECM by interacting with a variety of molecules in the matrix[20,61], while the central domain of different VCAN isoforms is decorated with different GAGs. Because of the negatively charged sulfates or carboxyl groups, the GAG chains of VCAN are attractive to various positively charged molecules such as certain growth factors and cytokines[10], by which VCAN helps maintain their immobilized gradients and offers protection for them from proteolytic cleavage[46]. These functions of VCAN including ECM integrity-maintaining effects will be lost upon proteolytic cleavage by ADAMTS4, leading to ECM remodeling and SROM at parturition. This notion was supported by observation of the amnion with TEM, which showed a distorted ECM structure following ADAMTS4 treatment in both in vitro human and in vivo mouse studies. Of interest, inhibition of ADAMTS4 has been shown to decrease rather than increase collagen content indirectly through modulation of the effect of TGF-β in the heart[62]. However, we failed to observe any obvious alterations in the total collagen abundance in the human amnion upon ADAMTS4 treatment in this study, suggesting that VCAN degradation rather than reduction in collagen abundance may contribute to ADAMTS4-induced ECM remodeling in the amnion.

Although 6 members (ADAMTS1, 4, 5, 9, 15, and 20) of the ADAMTS family have been identified to carry proteoglycanase activity toward VCAN[25,26], our transcriptomic sequencing data showed that ADAMTS4 was the only member manifesting a significant increase in transcription in the human amnion in deliveries with SROM. In addition to VCAN, ADAMTS4 is also capable of cleaving all the other members of the hyalectan class of chondroitin sulfate proteoglycan, including ACAN, BCAN, NCAN[63,64]. As our study revealed VCAN is the predominant proteoglycan of the hyalectan class in the amnion, we believe that the observed effect of intra-amniotic administration of ADAMTS4 on parturition in the mouse is derived, at least in part, from its proteolytic effect on VCAN. Increased VCAN degradation was indeed observed both in ADAMTS4-treated human amnion tissue explants and in the mouse fetal membranes following intra-amniotic administration of lentiviral vector expressing *Adamts4* in the present study.

The negatively charged sulfates or carboxyl groups of the GAG chains of VCAN are able to attract various positively-charged molecules including water and growth factors as well as proinflammatory cytokines and chemokines[10]. VCAN degradation will reduce this attraction resulting not only in a deformed ECM structure in the amnion, but also enhanced inflammatory reactions of the fetal membranes through mobilization of those cytokines and chemokines attracted to the GAG chains. In addition, versikine cleaved from VCAN by ADAMTS4 may also serve as an ECM-derived damage-associated molecular pattern to evoke inflammation in the fetal membranes. In the present study, we found that the expression of proinflammatory mediators including cytokines, chemokines and COX-2 were significantly enhanced by versikine in human amnion fibroblasts as well as in the mouse fetal membranes following intra-amniotic administration of lentiviral vector expressing *Adamts4*. Our findings are supported by the proinflammatory actions of versikine in non-gestational tissue[52–54]. Increased COX-2 expression will lead to enhanced prostaglandin synthesis, particularly PGE2 and PGF2α, the two eicosanoids that are crucial for cervical ripening and myometrial contraction in parturition[36]. In consideration of the amnion being the largest supplier of PGE2 in pregnancy, this action of versikine is no doubt another important contributing factor to ADAMTS4-induced preterm birth. Notably, we found in this study that the major proinflammatory factors involved in the inflammatory reactions of the fetal membranes such as IL-1β and SAA1 were capable of inducing ADAMTS4 expression and reducing its endocytosis in amnion fibroblasts, which is in line with previous studies showing that proinflammatory cytokines were capable of inducing ADAMTS expression and LRP1 shedding in osteoarthritis and lung inflammation[65,66]. In this regard, inflammation of the fetal membranes and ADAMTS4-mediated VCAN degradation may form a mutual reinforcing cycle in the amnion at parturition, which fits well into the feedforward mechanism of parturition.

Unlike the predominant expression of VCAN in the amnion, ACAN is the predominant hyalectan isoform in joint cartilage, with ADAMTS4

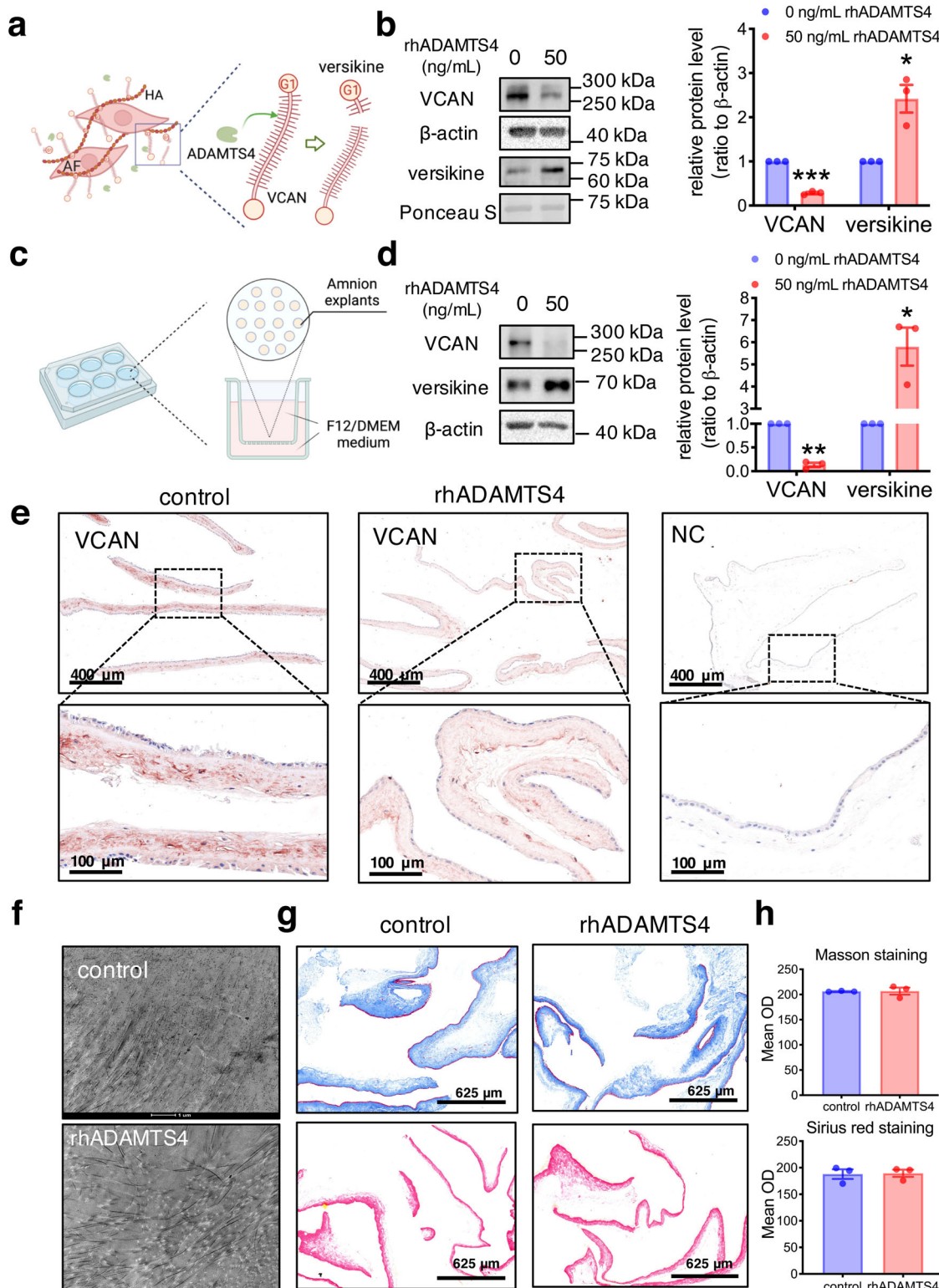

**Fig. 3 | Effect of ADAMTS4 on VCAN degradation and ECM remodeling in the human amnion. a** The illustration showing the presence of VCAN on the cell surface attached to hyaluronan and the cleavage of VCAN to versikine by ADAMTS4. AF, amnion fibroblasts; HA, hyaluronan; G1, N-terminal G1 Domain of VCAN. **b** Treatment of amnion fibroblasts with rhADAMTS4 (50 ng/mL; 24 h) decreased VCAN abundance in the cells, and increased versikine abundance in the conditioned culture medium ($n = 3$). rhADAMTS4, recombinant human ADAMTS4. **c** The illustration showing human amnion tissue explant culture. **d** rhADAMTS4 (50 ng/mL; 24 h) treatment of amnion tissue explants increased VCAN cleavage to versikine ($n = 3$). **e** Representative immunohistochemistry images showing decreased VCAN protein abundance in human

amnion tissue explants with rhADAMTS4 (50 ng/mL; 24 h) treatment ($n = 3$). Scale bars, 100 or 400 μm. **f** Representative images of transmission electron microscopy showing distorted collagen fibrils in the ECM of amnion tissue explants treated with rhADAMTS4 (50 ng/mL; 24 h). $n = 3$. Scale bars, 1 μm. **g** Representative images of Masson trichrome staining (blue) and Sirius red staining (red) showing unaltered total collagen abundance in amnion tissue explants treated with rhADAMTS4 (50 ng/ml; 24 h; $n = 3$), Scale bars, 625 μm.; (**h**). The average data of Masson staining and Sirius red staining images. OD, optical density. Data are mean ± SEM. Statistical analysis was performed with paired Student's $t$ test. *$p < 0.05$, **$p < 0.01$, ***$p < 0.001$ vs. control.

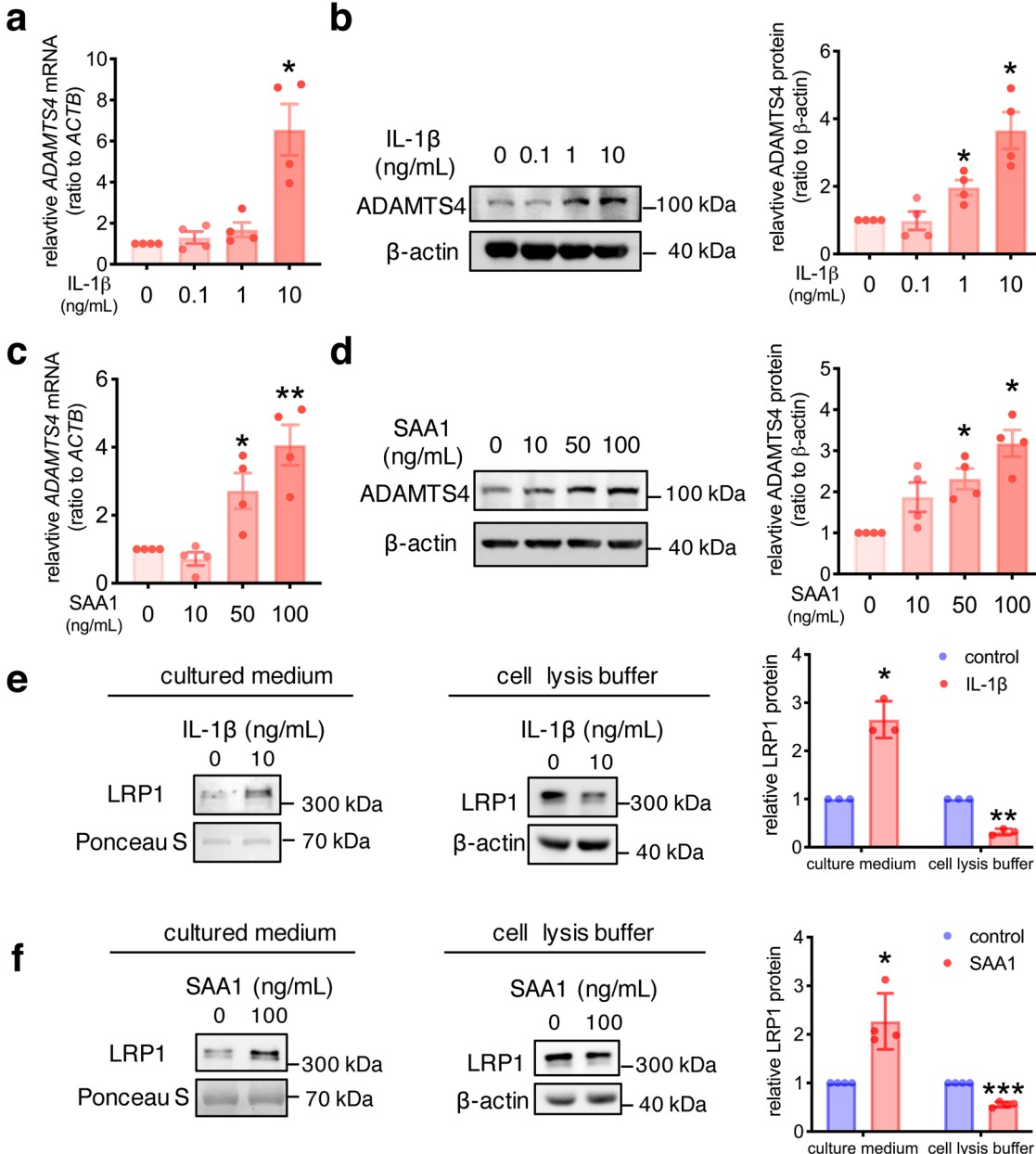

**Fig. 4 | Regulation of ADAMTS4 expression and LRP1 by IL-1β and SAA1 in human amnion fibroblasts. a–d** Concentration-dependent induction of ADAMTS4 mRNA ($n = 4$, a, c) and protein ($n = 4$, b, d) expression by IL-1β (0, 0.1, 1 and 10 ng/mL; 24 h) and SAA1 (0, 10, 50 and 100 ng/mL; 24 h) in human amnion fibroblasts. **e** and **f** Induction of LRP1 shedding by IL-1β (10 ng/mL; 24 h; $n = 3$; e) and SAA1 (100 ng/mL; 24 h; $n = 4$; f) in human amnion fibroblasts, illustrated by decreased LRP1 in the cells and increased LRP1 in the conditioned culture medium. Data are mean ± SEM. Statistical analysis was performed with one-way ANOVA test followed by Newman-Keuls multiple-comparisons test (**a−d**) or paired Student's *t* test (**e, f**). *$p < 0.05$, **$p < 0.01$, ***$p < 0.001$ vs. control (0).

as its principal cleaving enzyme as well[67,68]. With increasing age, ACAN abundance is greatly depleted, which is believed to be ascribed to the enhanced degradation by ADAMTS4 due to altered ACAN glycosylation[69]. Hence, ADAMTS4 is becoming an attractive therapeutic target for treatment of age-related osteoarthritis. Interestingly, the fetal membranes also undergo senescence with increasing gestational age[70,71]. Thus, an interesting issue arises whether glycosylation of VCAN in the amnion is also altered with gestational age resulting in enhanced VCAN cleavage by ADAMTS4 so that the membrane tensible strength can be weakened for rupture at parturition. In answer to this question, increasing VCAN cleavage was indeed observed in the mouse fetal membranes with gestational age in this study, suggesting that it is likely that glycosylation of VCAN is altered with gestational age although increasing ADAMTS expression may also be a factor.

In conclusion, we have provided evidence that ADAMTS4 plays a crucial role in parturition, and ADAMTS4-mediated VCAN cleavage is possibly one of the underlying mechanisms. Inflammation of the fetal membranes and ADAMTS4-mediated VCAN degradation appear to be a part of the reciprocal causative relationship at parturition. Given the role of ADAMTS4 illustrated in this study, we propose that ADAMTS4 in the amnion might serve as an effective therapeutic target for the prevention of PPROM and preterm birth in the future.

## Methods
### Human amnion collection
Human fetal membranes were collected with written informed consent from normotensive pregnant women at term (37–40 weeks) following TL group and TNL group under a protocol approved by the Ethics

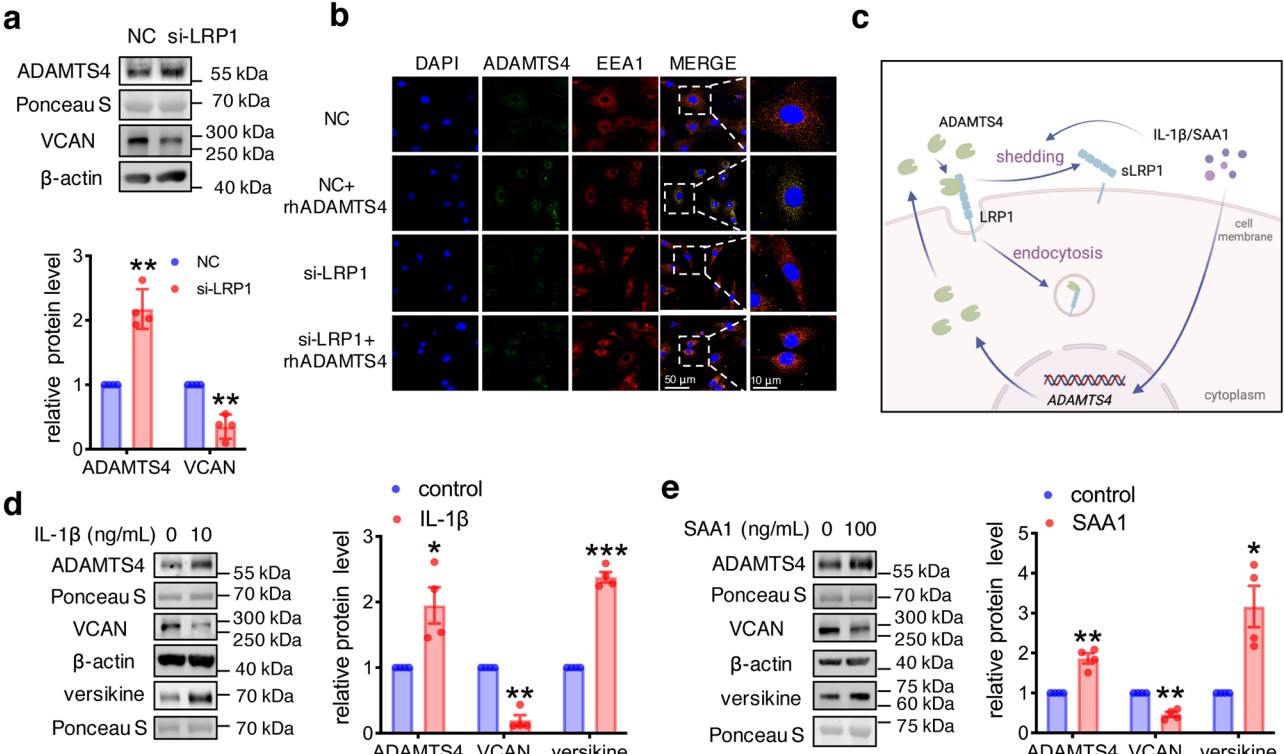

**Fig. 5 | Role of LRP1 in the endocytosis of ADAMTS4 in human amnion fibroblasts. a** Increased ADAMTS4 abundance in conditioned culture medium (n = 4), and decreased VCAN abundance in amnion fibroblasts (n = 4) following siRNA-mediated knockdown of LRP1. **b** Representative confocal microscopic images showing ADAMTS4 endocytosis illustrated by co-localization of ADAMTS4 (green) and EEA1 (red), an early endocytosis marker, in the endosomes of human amnion fibroblasts treated with rhADAMTS4 (20 nM; 6 h). ADAMTS4 endocytosis was blocked by siRNA-mediated knockdown of *LRP1* illustrated by the absence of ADAMTS4 in endosomes. n = 3. EEA1, early endosome antigen 1. NC, negative

control. Scale bar, 10 or 50 μm. **c** Diagram illustrating the inhibitory effects of IL-1β and SAA1 on ADAMTS4 endocytosis in human amnion fibroblasts. IL-1β and SAA1 not only induced ADAMTS4 expression but also inhibited ADAMTS4 endocytosis by enhancing LRP1 shedding from the cell membranes. sLRP1, shedding LRP1. **d** and **e** Treatment of human amnion fibroblasts with IL-1β (10 ng/mL; 24 h; d) and SAA1 (100 ng/mL; 24 h; e) increased ADAMTS4 abundance in the conditioned culture medium and decreased VCAN abundance in the cells (n = 4). Data are mean ± SEM. Statistical analysis was performed with paired Student's t-test. *p < 0.05, **p < 0.01, ***p < 0.001 vs. control (0) or NC.

Committee of Ren Ji Hospital, School of Medicine, Shanghai Jiao Tong University (approval code: RA-2022-172). All ethical regulations relevant to human research participants were followed. The amnion layer was peeled off the fetal membranes immediately after delivery for further processing. Pregnancies with complications such as gestational diabetes, preeclampsia, and fetal growth restriction were not included in this study.

## Isolation and culture of human amnion fibroblast and epithelial cells

To avoid the confounding effect of labor, the entire amnion of the reflected fetal membranes only from the TNL group was used for isolation of amnion fibroblasts. Briefly, the amnion tissue was digested twice with 0.125% trypsin (Life Technologies Inc., Grand Island, NY) at 37 °C for 20 min, and then washed thoroughly with phosphate buffered saline (PBS) for isolation of epithelial cells. The epithelial cells in trypsin-digested medium were collected by centrifugation. The remaining mesenchymal tissue was further digested with 0.1% collagenase (Sigma, St. Louis, MO) for 20 min to release fibroblasts. The isolated fibroblasts in the digestion medium were then spun down and resuspended in Dulbecco's Modified Eagle Medium (DMEM) containing 10% fetal bovine serum (FBS) and antibiotics (Life Technologies Inc.), and cultured in a 6-well culture plate at a density of 1 x 10⁶ cells/mL at 37 °C in 5% CO₂/95% air. High purity of fibroblasts was achieved. Staining with cytokeratin-7, vimentin and CD45, the respective markers for epithelial, mesenchymal and immune cells, showed that >95% of cells are fibroblasts[72].

## Treatment of human amnion fibroblasts

Three days after plating, the culture medium of amnion fibroblasts was replaced with DMEM containing 1% antibiotics but without phenol red and FBS prior to treatment with reagents. To study the effect of ADAMTS4 on VCAN degradation, fibroblasts were treated with rhA-DAMTS4 (50 ng/mL, R&D systems, Minneapolis, MN; #4307-AD-020) for 24 h. After treatment, fibroblasts were collected for measurement of cellular VCAN, and the conditioned culture medium was collected for measurement of versikine with Western blotting after concentration. To examine the regulation of ADAMTS4 and LRP1 by proinflammatory factors, fibroblasts were treated with IL-1β (0, 0.1, 1, 10 ng/mL; Sigma) and SAA1 (0, 10, 50, 100 ng/mL; PeproTech Inc., Rocky Hill, NJ) for 24 h. To explore the role of ADAMTS4 in VCAN degradation by IL-1β, fibroblasts were treated with IL-1β (10 ng/mL, Sigma) in the presence or absence of an ADAMTS4 antagonist, an analogue peptide of TSP-1 motif of ADAMTS4 (ᴺRGGWGPWGPWGDCSRTCGGGᶜ (s-s bound))[50,51] (10 μM, Hongtide Co. Ltd, Shanghai, China). After treatments, fibroblasts were collected for determination of cellular ADAMTS4, LRP1 and VCAN with Western blotting and qRT-PCR, and the conditioned culture medium was collected for measurement of secreted ADAMTS4, versikine and shed LRP1 with Western blotting after concentration. The protein in the conditioned culture medium was concentrated with a 10 kDa centrifugal filter device (Millipore) according to the manufacturer's instruction.

To explore the involvement of LRP1 in the endocytosis of ADAMTS4, fibroblasts were treated with or without rhADAMTS4 (20 nM; R&D systems) in the presence or absence of siRNA-mediated knockdown of LRP1. The

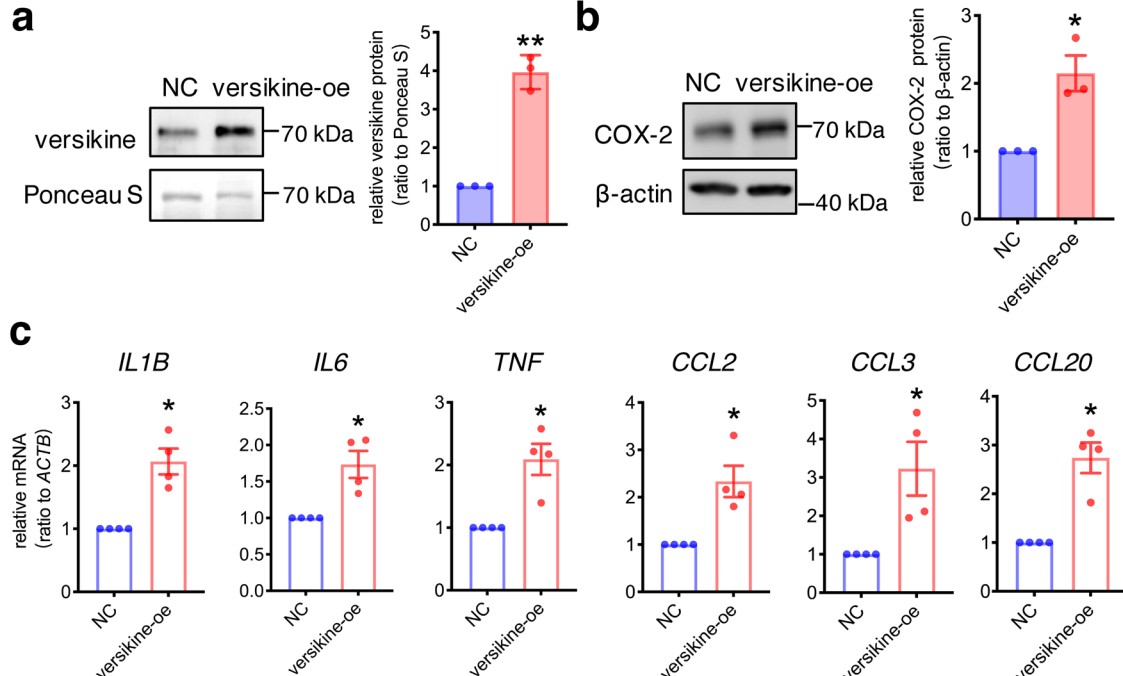

**Fig. 6 | Stimulation of proinflammatory factors by versikine in human amnion fibroblasts. a** Western blotting analysis showing increased secretion of versikine in the conditioned culture medium of amnion fibroblasts transfected with vectors expressing versikine ($n = 3$). NC, negative control; versikine-oe, versikine over-expression. **b** and **c** Induction of COX-2 protein (b; $n = 3$), proinflammatory cytokines (*IL1B, IL6, TNF*) and chemokines (*CCL2, CCL3, CCL20*) mRNA (c; $n = 4$) in amnion fibroblasts transfected with vectors expressing versikine. Data are mean ± SEM. Statistical analysis was performed with paired Student's *t* test. *$p < 0.05$, **$p < 0.01$ vs. NC.

culture medium was then collected for measurement of secreted ADAMTS4 with Western blotting after concentration. For siRNA transfection, fibroblasts were incubated with 50 nM siRNA (5'-GCGCAUCGAUCUUCA CAAATT-3') against *LRP1* or randomly scrambled siRNA (5'-UUCUCC GAACGUGUCACGUTT-3') (GenePharma, Shanghai, China) in Opti-MEM (Life Technologies Inc.) immediately after isolation using an electro-porator (165 V for 5 ms) (Nepa Gene, Chiba, Japan). The cells were then recovered in DMEM containing 10% FBS and 1% antibiotics for three days before treatment with rhADAMTS4 for 6 h. The knockdown efficiency of LRP1 was about 95% at the protein level (Supplementary Fig. S6).

The role of versikine in the inflammation reaction was studied by transfecting amnion fibroblasts with versikine-expressing plasmids using Lipofectamine® 3000 Transfection Kit (Invitrogen, San Diego, CA; #2563912) 3 h after isolation. Amnion fibroblasts were transfected with a mixture of lipo3000, P3000 and versikine-expressing plasmid in Opti-MEM (Life Technologies Inc.) for 12 h in DMEM containing 10% FBS. After transfection, the culture medium was replaced with DMEM containing 1% antibiotics but without phenol red and FBS followed by incubation for another 60 h. The overexpression efficiency of versikine, evaluated by measuring versikine in the conditioned culture medium with Western blotting after concentration, was about 400% at the protein level (Fig. 6a).

**Preparation and treatment of human amnion tissue explants**
Likewise, amnion tissue from the TNL group ($n = 6$) was used to avoid the confounding effect of labor. After washing off the residual blood with PBS, the amnion tissue was incised into 6-mm circles in diameter using a skin-biopsy punch. The amnion discs were then placed in Falcon cell-culture inserts (BD Biosciences, Franklin Lakes, NJ) and cultured in a 6-well plate in DMEM/F12 medium containing 10% FBS and 1% antibiotics (Life Technologies Inc.) in 5% $CO_2$/95% air at 37 °C for 24 h. After equilibration, the culture medium was replaced with fresh medium for treatment with rhADAMTS4 (50 ng/mL; R&D systems) for 24 h. After treatment, tissue explants were collected for measurement of VCAN and versikine with Western blotting or

immunohistochemical staining. Tissue explants were also stained with Masson trichrome and Sirius red for examination of total collagen abundance or processed for observation of ECM structure with TEM as follows.

**Immunohistochemical staining of human amnion tissue**
To examine the distribution of VCAN, versikine, ADAMTS4 and LRP1 in the human amnion, the amnion tissue from artificial (TNL) and spontaneous (TL) rupture sites was fixed in 4% paraformaldehyde and embedded in par-affin for sectioning at 3 μm thickness. After deparaffinization and quenching of endogenous peroxidase activity with 0.3% $H_2O_2$, non-specific binding sites were blocked by incubating the section with normal horse serum. Then pri-mary antibodies against human VCAN (1:100; R&D systems; #AF3054), versikine (1:200; Thermo Fisher scientific, Waltham, MA; #PA1-1748A), ADAMTS4 (1:100; abcam, Cambridge, U.K.; #ab185722), LRP1 (1:200; ABclonal, Wuhan, China; #A1439) were applied for incubation overnight at 4 °C. Non-immune serum (Proteintech, Wuhan, China) served as negative control. The section was washed and incubated with corresponding bioti-nylated secondary antibodies for 1 h, followed by incubation with avidin-biotin complex conjugated with horseradish peroxidase (Vector Laboratories, Burlingame, CA) for 30 min. The red color reaction was then developed using 3-amino-9-ethyl carbazole (Vector Laboratories) as the substrate for horse-radish peroxidase. The section was counterstained with hematoxylin (blue color) and examined under a regular light microscope (Zeiss, Oberkochen, Germany). For VCAN observation, sections were pre-treated with 0.4 μ/mL Chondroitinase ABC (Sigma; #C3367) in 50 mM Tris (pH 8.0), 60 mM sodium acetate, and 0.02% bovine serum albumin (BSA) for 1 h at 37 °C before incubation with VCAN antibody in order to remove glycosami-noglycan chains for better presentation of antigen to VCAN antibody.

**Masson trichrome and Sirius red staining of human amnion tissue**
For histological evaluation of total collagen abundance in the amnion tissue with and without rhADAMTS4 treatment, Masson trichrome and

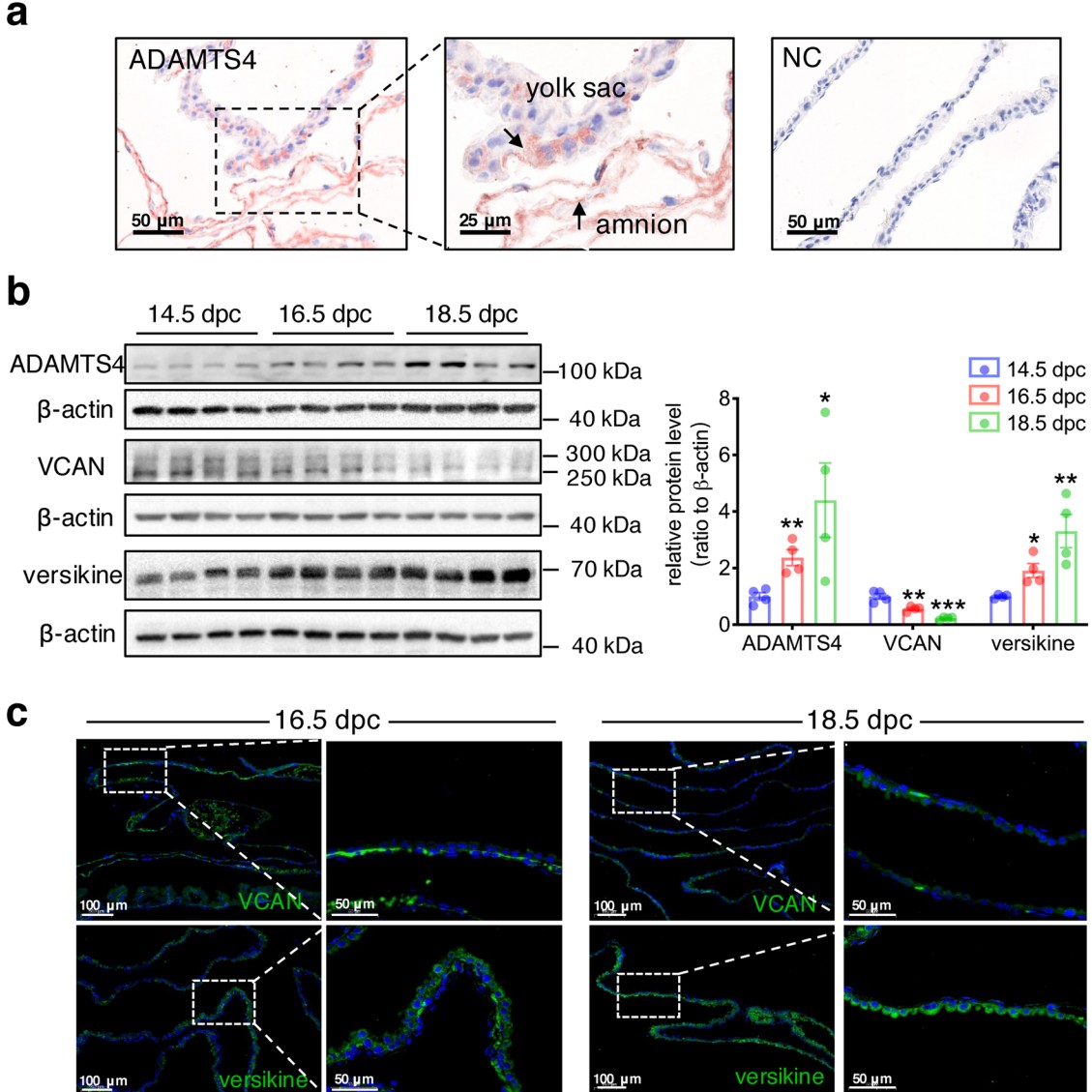

**Fig. 7 | Gestational age-dependent changes in ADAMTS4, VCAN and versikine abundance in mouse fetal membranes. a** Representative immunohistochemical images showing the presence of ADAMTS4 in both amnion and yolk sac layers of the mouse fetal membranes on 16.5 dpc. NC, negative control. dpc, day post-coitum. Scale bars, 25 or 50 μm. **b** Western blotting analysis showing gestational age-dependent alterations in ADAMTS, VCAN and versikine protein abundance in mouse fetal membranes from 14.5 to 18.5 dpc. **c** Representative images of immunofluorescence staining showing decreased VCAN (green) and increased versikine (green) abundance on 18.5 dpc as compared to 16.5 dpc in mouse fetal membranes (*n* = 3). Scale bars, 50 or 100 μm. Data are mean ± SEM. Statistical analysis was performed with one-way ANOVA test followed by Newman-Keuls multiple-comparisons test. *$p < 0.05$, **$p < 0.01$, ***$p < 0.001$ vs. 14.5 dpc.

Sirius red staining (Sbjbio Life Sciences, Nanjing, China; #190801/200405) were performed on sections prepared from paraffin-embedded amnion tissue explants after fixing in 4% paraformaldehyde using commercial kits following a protocol from the manufacturer. The section was then examined under bright field of a regular microscope (Zeiss). The blue (Masson trichrome) or red (Sirius red) staining areas representing total collagen abundance were measured from three random visual fields with ImageJ software (National Institute of Health, Bethesda, MD), and the measurements averaged to indicate the relative abundance of total collagen.

**TEM examination of human amnion tissue**
To observe the ECM structure of human amnion tissue explants with or without rhADAMTS4 (50 ng/mL; R&D systems; 24 h) treatment, the amnion tissue explants were fast fixed with 2.5% glutaraldehyde at 4 °C overnight followed by fixing in 1% osmium tetroxide for 2 h at 4 °C. After dehydration, infiltration, routine embedding and polymerization of the

fixed tissue, ultrathin tissue sections (70 nm) were cut and stained with lead citrate and uranium acetate for observation with a transmission electron microscope (Talos L120C; Thermo Fisher Scientific).

**Dual immunofluorescence staining in human amnion fibroblasts**
Dual immunofluorescence staining was performed to examine the colocalization of ADAMTS4 and EEA1, a well-characterized early endosomal marker[49], in human amnion fibroblasts. After treatment with or without rhADAMTS4 in the presence or absence of siRNA-mediated knockdown of LRP1, the cells were fixed in 4% paraformaldehyde for 20 min, followed by permeabilization with 0.4% Triton X-100 for 20 min. After blocking with normal goat serum, the cells were incubated with the antibody against human ADAMTS4 (Abcam; #ab185722) followed by incubation with the antibody against EEA1 (Abcam; #ab70521) at 1:100 dilution. After washing, incubation with secondary antibody conjugated with Alexa Fluor 488 (green color) for ADAMTS4 or Alexa Fluor 594 (red color) for EEA1 (both from Proteintech) was conducted sequentially. Nuclei were counterstained with

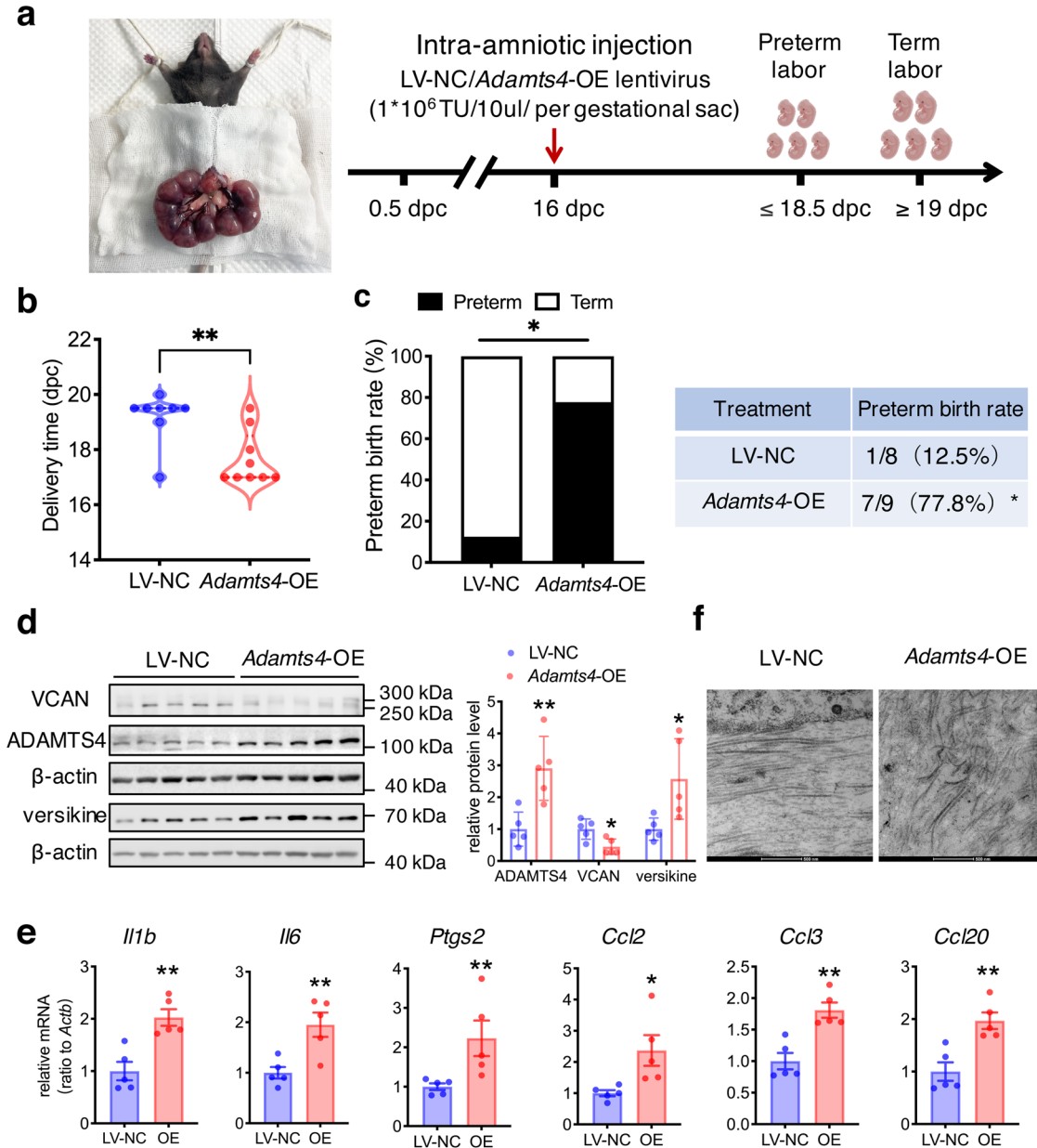

**Fig. 8 | Induction of preterm birth by intra-amniotic injection of lentiviral vector expressing ADAMTS4 in the mouse. a** Time-line illustrating the procedure of ADAMTS4 administration in pregnant mice. LV-NC, lentiviral vector-negative control. **b** and **c** Effects of intra-amniotic injection of l;entiviral vector expressing *Adamts4* (1 × 10⁶ TU/10 μl/ per gestational sac on 16 dpc; *n* = 9) on delivery time (b) and preterm birth rate (c). LV-NC mice were given an equal amount of lentiviral empty vector (*n* = 8). **d** and **e** Effect of intra-amniotic injection of lentiviral vector expressing *Adamts4* (1 × 10⁶ TU/10 μl/per gestational sac on 16 dpc) on the protein abundance of ADAMTS4, VCAN and versikine (d) and mRNA abundance of *Ptgs2* (encoding COX-2), proinflammatory cytokines (*Il1b, Il6*) and chemokines (*Ccl2, Ccl3, Ccl20*) (e). **f** Representative TEM images showing distorted collagen fibrils in the ECM of mouse fetal membranes following intra-amniotic injection of lentiviral vector expressing *Adamts4* (1 × 10⁶ TU/10 μl/ per gestational sac on 16 dpc). Scale bars, 500 nm. Data are mean ± SEM. Statistical analysis was performed with Mann–Whitney U test (**b**), Fisher exact test (**c**) or unpaired Student's *t* test (**d, e**). *$p < 0.05$, **$p < 0.01$ vs LV-NC.

DAPI (blue color). A confocal microscope system (Leica, Wetzlar, Germany) was used to observe immunofluorescence signals.

### Extraction of RNA and analysis with qRT-PCR

Total RNA was isolated from the amnion tissue of the TNL and TL groups or from treated amnion fibroblasts using a commercial kit (Foregene, Chengdu, China) according to the manufacturer's instruction. After examination of RNA concentration and quality using a Nanodrop (Thermo Fisher Scientific), reverse transcription was carried out using a Prime-Script RT Master Mix Kit (TaKaRa, Kyoto, Japan). The mRNA transcripts of *VCAN, ACAN, BCAN, NCAN, ADAMTS4, IL1, IL6, TNF, CCL2, CCL3, and CCL20* were determined with qRT-PCR using the above reverse-transcribed cDNA, power SYBR® Premix Ex Taq™ (TaKaRa) and primers illustrated in Supplementary Table S1. The housekeeping gene, *ACTB*, was amplified in parallel as an internal loading control. The relative expression levels were analyzed by the $2^{-\Delta\Delta ct}$ method.

### Extraction of protein and analysis with Western blotting

Total cellular protein was extracted from the snap-frozen amnion tissue of the TNL and TL groups or treated amnion fibroblasts with an ice-cold radioimmunoprecipitation assay (RIPA) lysis buffer (Active Motif, Carlsbad, CA) containing inhibitors for both protease and phosphatase (Roche, Indianapolis, IN). Secreted protein in the conditioned culture medium was collected and concentrated. After quantification of protein

**Article**

concentration with the Bradford method, 25 μg of protein from each sample was electrophoresed in 8 or 10% SDS–polyacrylamide gel, and then transferred to a nitrocellulose membrane blot. After blocking with 5% non-fat milk, the membrane blot was incubated with primary antibodies against VCAN (1:1000; abcam; #ab270445), versikine (1:1000; Thermo Fisher Scientific; #PA1-1748A), ADAMTS4 (1:1000; abcam; #ab185722 (cellular) or Proteintech; #11865-1-AP (secreted)), LRP1 (1:1000; ABclonal #A1439), COX-2 (1:1000; Cell Signaling Technology, Danvers, MA; #12282), Vimentin (1:20000; abcam; #ab92547), E-cadherin (1:2000; Cell Signaling Technology; #3195) overnight at 4 °C, followed by incubation with corresponding secondary antibodies conjugated with horseradish peroxidase. The peroxidase activity was developed with a chemiluminescence detection system (Millipore, Billerica, MA), and visualized using a G-Box chemiluminescence image capture system (Syngene, Cambridge, U.K.). Internal loading control was performed by either probing the same blot with an antibody against β-actin (1:10000; Proteintech; #60008-1) for cellular protein or staining the same blot with Ponceau S for secreted protein in the conditioned culture medium. The ratio of band intensities of VCAN, versikine, ADAMTS4, LRP1 and COX-2 to that of β-actin or Ponceau S staining was used to indicate the target protein abundance. For the measurement of VCAN, total protein was pre-digested with 0.4 μ/mL Chondroitinase ABC (Sigma; #C3367) in 50 mM Tris (pH 8.0), 60 mM sodium acetate, and 0.02% BSA for 1 h at 37 °C before electrophoresis in order to remove glycosaminoglycan chains and improve antibody binding.

### Animal study

The mouse study was carried out following ARRIVE guidelines and followed the relevant ethical regulations for animal use, which was approved by the Institutional Review Board of Ren Ji Hospital, Shanghai Jiao Tong University School of Medicine. C57BL/6 mice (Charles River, Beijing, China) of 10 to 13 weeks age were mated overnight. Gestational day was counted as 0.5 dpc when a vaginal plug was present.

To study gestational age-dependent changes in ADAMTS4, VCAN, versikine abundance, mouse fetal membranes were collected on 14.5, 16.5 and 18.5 dpc. The abundance and distribution of ADAMTS4, VCAN and versikine in the fetal membranes was determined with Western blotting, immunohistochemical and immunofluorescence staining using antibodies against mouse ADAMTS4 (1:1000; abcam; #ab185722), VCAN (1:1000; ABclonal; #A19655) and versikine (1:1000; Thermo Fisher Scientific; #PA1-1748A).

To examine whether ADAMTS4 was able to cleave VCAN resulting in enhanced expression of proinflammatory factors and ECM remodeling in the fetal membranes thereby leading to preterm birth, intra-amniotic injection of lentiviral vectors expressing *Adamts4* ($1 \times 10^6$ TU/10 μl/per gestational sac; Genechem, Shanghai, China) was performed on 16 dpc under laparotomy after anesthesia with 65 mg/kg Zoletil 50 (Virbac, Nice, France). The same amount of empty lentivirus vector served as negative control. Some of the mice ($n = 17$) were allowed to deliver spontaneously for observation of delivery time, and some ($n = 10$) were sacrificed 18 h after surgery for collection of fetal membranes to examine the abundance of ADAMTS4, VCAN and versikine with Western blotting and immunofluorescence staining. The expression of *Il1, Il6, Ccl2, Ccl3, Ccl20, Ptgs2* mRNA in the fetal membranes was also measured with qRT-PCR with the primers illustrated in Supplementary Table S1. TEM was used to observe ECM structure of the fetal membranes. The lentiviral vector expressing mouse *Adamts4* gene was designed and constructed by Shanghai Gene-Chem, Co. Ltd, China. Briefly, three plasmids (plasmid Helper1.0, plasmid Helper2.0 and the plasmid expressing *Adamts4*) were co-transfected into HEK293T cells with transfection reagent (Genechem, Shanghai, China) for 48−72 h. After transfection, the culture medium of HEK293T cells was collected and centrifuged for virus purification. The quality of lentiviral vectors expressing *Adamts4* was monitored by physical state, sterility and virus titer testing.

### Statistics and reproducibility

All the experiments were performed at least three times independently. The n number in each experiment represents independent experiments using samples from individual pregnant women or mice. All data are presented as mean ± SEM. Shapiro-Wilk test was performed to examine the normality of the data. Based on the normality test, paired or unpaired Student's *t* test or Mann-Whitney U-test was used to compare two groups where appropriate. One-way ANOVA followed by Newman-Keuls multiple-comparisons test was performed when assessing the normally distributed data among three or more groups. Statistical significance was defined as $P < 0.05$.

### Reporting summary

Further information on research design is available in the Nature Portfolio Reporting Summary linked to this article.

### Data availability

Supplementary Fig. S7-24 contains the original uncropped blot/gel images of the main figures and Supplementary Figs. Supplementary Data 1 contains the source data for the graphs in the figures. The other original data presented in the study can be directed to the corresponding authors.

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

## Acknowledgements

This study was supported by National Key R & D Program of China (2022YFC2704602), National Natural Science Foundation of China (81830042, 82071677 and 82271717); and Innovative Research Team of High-level Local Universities in Shanghai (SHSMU-ZLCX20210201); and Three-Year Action Plan for Strengthening the Construction of the Public Health System in Shanghai (GWVI-11.1-36); and Shanghai's Top Priority Research Center Construction Project (2023ZZ02002). We are grateful to Lu-yao Wang and Hao Ying for assistance with sample collection, and to the staff in the Electron Microscopy center of Shanghai Institute of Precision Medicine, Shanghai Ninth People's Hospital, Shanghai Jiao Tong University School of Medicine, for their assistance in the study with the transmission electron microscopy.

## Author contributions

K.S. and W.-S.W. conceptualization, supervision and funding acquisition; M.-D.L., J.-W.L., F.Z., W.-J.L., F.P., Y.-K.L. and L.-J.L investigation and methodology; M.-D.L., W.-S.W., L.M. and K. S. writing.

## Competing interests

The authors declare no competing interests.
