## [Peer Review File · Communications Biology]

Reviewers' comments:

Reviewer #1 (Remarks to the Author):

The manuscript entitled “The non-fibrous versican cleavage by ADAMTS4 is crucial for fetal membrane rupture at parturition” is an interesting and thorough investigation of how ADAMTS4 increases in abundance, how the endocytic receptor for ADAMTS4 is gradually downregulated, and how ADAMTS4 cleaves versican into versikine, which in turn acts as a DAMP to induce inflammation in the fetal membranes. I have some minor comments and one concern that data in one panel is mislabeled. I do feel that the experiments performed do not completely support the title of the manuscript, particularly that versikine is crucial for fetal membrane rupture at parturition. The mechanistic animal experiments showed that ADAMTS4 overexpression resulted in preterm birth. It is a logical but experimentally unsupported leap to say that versikine is crucial for membrane rupture at parturition. If the authors were to instead inhibit ADAMTS4 in a mouse model and show that decreases in versikine (due to decreased versican cleavage) corresponded to delayed rupture of membranes, and that overexpressing or supplementing the model with versikine rescued this, *that* would show that versikine is crucial for membrane rupture at parturition. A title directly supported by the data in this manuscript would be something more like “ADAMTS4 activity and abundance partially controls fetal membrane rupture”.

Intro, lines 97-100: This sentence is inaccurate. Chorioamnionitis is any inflammation of the fetal membranes, not strictly infectious. Chorioamnionitis can be sterile or infectious.

Figure S1B: while it's understandable that the authors only did RT-PCR on ACAN, because it was the only one that showed up on large-scale transcriptomics sequencing, it would be more complete to include qRT-PCR for NCAN and BCAN as well to ensure that the transcriptomics data was complete and accurate.

Figure 2A: the authors state that ADAMTS4 is the only ADAMTS family member showing significant increase in transcript level during parturition; however, ADAMTSs 9 and 1, while not significantly induced by transcriptomic sequencing, may have been increased as well if checked by qRT-PCR and may merit investigation. Each of ADAMTS 9 and 1 have what looks like one low value outlier out of 3 values, the other two appearing high. This doesn't mean that ADAMTS4 isn't responsible for most or all of the cleavage, but that there may be some redundancy and that ADAMTSs 9 and 1 also contribute. LRP1 also binds other ADAMTSs including ADAMTS9.

Figure 2F and 2G: the mRNA data from 2F does not appear to correspond to the protein data in 2G. The blue bars from 2F show roughly equal expression of ADAMTS4 and LRP1 in the amnion epithelium at the mRNA levels, and considerably higher expression of ADAMTS4 than LRP1 in the amnion fibroblasts; the protein data from 2G shows extremely high amounts of LRP1 in the amnion epithelium and roughly equal amounts of LRP1 and ADAMTS4 in the amnion fibroblasts. This seems like much more of a difference between mRNA and protein expression than I'm used to seeing, and likely unaccounted for by housekeeping genes. All samples are taken from TNL amnion, including the AE and AF in vitro cell

culture. Are the data labels in 2F correct? It would make more sense if AE and AF data were swapped AND the ADAMTS4 and LRP1 data were also swapped. If all the data is accurate, please comment on the large discrepancy between the mRNA and protein data in the results or discussion.

Reviewer #2 (Remarks to the Author):

Title: The non-fibrous versican cleavage by ADAMTS4 is crucial for fetal membrane rupture at parturition

This study investigated the function of versican which is cleaved by ADAMTS4 in the rupture of fetal membranes at parturition.

The authors showed that versican is expressed in the human amnion and its expression was decreased in the amnion of term in labor (TL) compared to that of term not in labor (TNL) during. In contrast, versican's cleaved product, versikine in the amnion is increased in term in labor. Both versican and versikine were localized in amnion fibroblasts. Next, the expression of ADAMTS4 was analyzed in amnion and it was higher in TL than that of TNL. In contrast, the cell surface receptor of endocytosis, LRP1 expression, was lower in TL than that of TNL. By using single cell RNA seq and immunohistochemistry, the authors showed that both ADAMTS4 and LRP1 were highly expressed in amnion fibroblasts compared to amnion epithelial cells.

To test the effect of ADAMTS4 on the cleavage of versican, amnion fibroblasts were treated with recombinant ADAMTS4. As expected, treatment of ADAMTS4 decreased the versican, whereas the cleavage product versikine expression was reciprocally increased. This reciprocal effect by ADAMTS4 was reproduced in the organ culture of amnion tissues. TEM observation revealed that collagen fibrils were distorted by ADAMTS4 but total collagen amount was not changed, showing that protease ADAMTS4 can cleave versican and produce versikine.

Next, the effects of inflammatory cytokines, IL-1 β and SAA1 were examined. Both cytokines dose-dependently increased ADAMTS4 expression. The cytokines stimulated the release of LRP1 in the media and resulted in the decrease of cellular LRP1. Knockdown of LRP1 by siRNA increased the expression of ADAMTS4 and decreased that of versican, suggesting that blocking endocytosis of ADAMTS4 surely increased the released ADAMTS4 which cleaved and decreased free versican. Furthermore, transfection of versikine into amnion fibroblasts increased mRNA expression of inflammatory cytokines, and COX-2 protein.

In the pregnant mice, the expression of ADAMTS4 and versikine in the fetal membranes increased toward term whereas that of versican decreased. This expression change during pregnancy and labor is the same as observed in human amnion. Finally, induction of ADAMTS4 by lentivirus in the fetal membranes increased the preterm birth rate almost 6-fold. As expected, this model showed the

reciprocal change of versican and versikine and the increase of inflammatory cytokines. TEM revealed the distortion of collagen fibrils in the fetal membranes by forced expression of ADAMTS4. The authors beautifully showed that ADAMTS4-mediated versican degradation amplifies the cytokine release and degrades collagen fibrils during labor in the amnion by the sophisticated experiments using primary human amnion cells and pregnant mouse model. The manuscript is well-written. However, there are some concerns in the experiments of Figure 5B and 8, so major revision is necessary.

Major Comments:

- 1) Line 191. Please explain the background and the roles of serum amyloid A1 (SAA1) in relation to the membrane rupture and labor because this molecule is not familiar to readers.
- 2) Lines 202-204. The experiment of Figure 5B and its interpretation is confusing. This Figure 5B presumably wants to show that endocytosis of ADAMTS4 does not occur if LRP is knocked down. First, the reason why recombinant ADAMTS4 was used is unclear (Figure 5B, 2nd and 4th rows). I think the authors should use cytokines (IL-1 β) instead to induce ADAMTS4. Immunofluorescence of ADAMTS4 in the cells treated with recombinant ADAMTS4 is nonsense. So I recommend using IL-1 β instead of recombinant ADAMTS4 in this knockdown experiment and immunofluorescence. Second, was EEA1 expression decreased by si-LRP1? This is because the expression change of EEA1 between NC (1st row) and si-LRP1 (3rd row) is unclear from this image. Please explain the interpretation of the Figure 5B experiment.
- 3) Lines 212-214. The authors suggest the reciprocal expression change of versican and versikine by IL-1 β and SAA1 is mediated by ADAMTS4. If the authors want to prove the importance of ADAMTS4, an additional experiment might be necessary that amnion fibroblasts are treated by cytokines under the existence of ADAMTS4 inhibitor and see the expression of versican and versikine.
- 4) Lines 234-237. The results of the experiments of intra-amniotic injection of a lentiviral vector expressing Adamts4 (Fig. 8) should be carefully interpreted, although the experiment itself is sophisticated. This preterm birth was induced by forced expression of Adamts4, not by versican-versikine. Therefore, it is possible that preterm birth and concomitant cytokine increase was induced by other molecules induced by Adamts4 than versican-versikine. Proteases such as ADAMTS4 have potency to directly induce cytokines which might have caused preterm birth in this model. In other words, this experiment does not prove the direct versican-versikine effect to pregnant mice. Why did not the authors inject versikine itself into mice instead of ADAMTS4? This experiment alone cannot lead to the conclusion as “the observed effect of intra-amniotic administration of ADAMTS4 on parturition in the mouse is derived largely from its proteolytic effect on VCAN” (lines 275-278).

Minor comments:

- 1) Line 43. Abbreviations are typically defined the first time the term is used. It might be better to change “VCAN” here to “versican”.

2) Line 53. The term “preterm premature rupture of the membranes” was recently changed to “preterm prelabor rupture of the membranes”. It might be better to change “premature” to “prelabor”.

Reviewer #3 (Remarks to the Author):

1. Please specify the amnion cell type used (epithelial or mesenchymal) for the study in the abstract.
2. The rationale for choosing Hyalectan, from a mixture of collagen and non-collagen-rich extra cellular matrix (ECM) proteins, must be highlighted. This may include a breakdown of the % of hyalectan family members in the membrane matrix as they seem minimal. This is critical to support the hypothesis.
3. Also, a description of how they may impact membrane matrix architecture is also important. Inflammation often seems to drive the production of matrix proteins as an innate mechanism to maintain homeostasis. Please include a short description on how this will impact the hypothesis.
4. Fig 1. Of the nine samples, maybe few (~3) samples show increased versikine and the others are somewhat similar to TNL. Specifically, one outlier maybe driving the stats. Please comment and may downplay the role of versikine.
5. Fig 1D: A higher magnification image would help to determine the staining pattern in the matrix region. Looks like some nonspecific staining in the matrix region in versikine.
6. Fig 2: Since fibroblast of the amnion are increased during TL, do authors consider the increase in ADAMTS4 is a remnant of an expected increased fibroblast? This can also be an argument that authors should do that the increased fibroblasts and subsequent increase in ADAMTS4 is a physiologic response required at term to facilitate degradation of VCAN. May discuss this.
7. The rationale for choosing fibroblast for the studies needs explanation. Fig 1D shows a major shift in versikine in amnion epithelial layer and VCAN is scattered throughout the matrix (as expected) and not necessarily in the fibroblast. I would have chosen epithelial cells to study these factors or both cell types.
8. Transitioning amnion epithelial cells maybe a major and better source of versikine and that may give a good indication of how cellular transitions help to promote localized inflammation (as authors rightfully claimed) and promote ECM degradation.
9. Fig 4: Having the mRNA and proteins on the same scale does not make much sense. Please explain how normalization was done.
10. Fig 4: The western blot panels don't show any changes with IL-1b and SAA, although there is a significant change. May explain how the normalization was done.
11. The data on endocytosis of ADAMTS4 is interesting

Reviewer #1: The manuscript entitled “The non-fibrous versican cleavage by ADAMTS4 is crucial for fetal membrane rupture at parturition” is an interesting and thorough investigation of how ADAMTS4 increases in abundance, how the endocytic receptor for ADAMTS4 is gradually downregulated, and how ADAMTS4 cleaves versican into versikine, which in turn acts as a DAMP to induce inflammation in the fetal membranes. I have some minor comments and one concern that data in one panel is mislabeled. I do feel that the experiments performed do not completely support the title of the manuscript, particularly that versikine is crucial for fetal membrane rupture at parturition. The mechanistic animal experiments showed that ADAMTS4 overexpression resulted in preterm birth. It is a logical but experimentally unsupported leap to say that versikine is crucial for membrane rupture at parturition. If the authors were to instead inhibit ADAMTS4 in a mouse model and show that decreases in versikine (due to decreased versican cleavage) corresponded to delayed rupture of membranes, and that overexpressing or supplementing the model with versikine rescued this, *that* would show that versikine is crucial for membrane rupture at parturition. A title directly supported by the data in this manuscript would be something more like “ADAMTS4 activity and abundance partially controls fetal membrane rupture”.

Reply: Thanks for reviewing our manuscript and all the constructive comments. We agree that at the current stage, we can only conclude that ADAMTS4 plays a crucial role in parturition, and ADAMTS4-mediated versican cleavage is possibly one of the underlying mechanisms since there may be other unrevealed ADAMTS4-mediated effects in parturition. Following your suggestion, we have modified the title of the manuscript as well as the abstract and conclusion in the revised manuscript.

1. Intro, lines 97-100: This sentence is inaccurate. Chorioamnionitis is any inflammation of the fetal membranes, not strictly infectious. Chorioamnionitis can be sterile or infectious.

Reply: We are sorry for the inaccurate description here. As you said, chorioamnionitis can be sterile or infectious although chorioamnionitis is often referred to infectious inflammation of the fetal membranes clinically. To be more precise, we have changed the term “chorioamnionitis” to “infectious chorioamnionitis” where appropriate to indicate infectious inflammation of the fetal membranes throughout the text.

2. Figure S1B: while it’s understandable that the authors only did RT-PCR on ACAN, because it was the only one that showed up on large-scale transcriptomics sequencing, it would be more complete to include qRT-PCR for NCAN and BCAN as well to ensure that the transcriptomics data was complete and accurate.

Reply: Thank you for raising this issue. Following your suggestion, we measured BCAN and NCAN mRNA expression with qRT-PCR, and found that BCAN and NCAN were hardly detectable in human amnion tissue, which is consistent with our transcriptomic data. Please see the data in Figure S1B in the revised manuscript.

3. Figure 2A: the authors state that ADAMTS4 is the only ADAMTS family member showing significant increase in transcript level during parturition; however, ADAMTSs 9 and 1, while not significantly induced by transcriptomic sequencing, may have been increased as well if checked by qRT-PCR and may merit investigation. Each of ADAMTS 9 and 1 have

what looks like one low value outlier out of 3 values, the other two appearing high. This doesn't mean that ADAMTS4 isn't responsible for most or all of the cleavage, but that there may be some redundancy and that ADAMTSs 9 and 1 also contribute. LRP1 also binds other ADAMTSs including ADAMTS9.

Reply: This is an excellent point. As a matter of fact, by using qRT-PCR and Western blotting, we have also found that ADAMTS9 expression was upregulated in the human amnion obtained from spontaneous rupture of fetal membrane at parturition. In addition, we have also found that the regulation of ADAMTS9 is quite different from that of ADAMTS4 in amnion fibroblasts. The study on ADAMTS9 is currently undergoing in our laboratory. That is why this manuscript mainly focused on ADAMTS4.

4. Figure 2F and 2G: the mRNA data from 2F does not appear to correspond to the protein data in 2G. The blue bars from 2F show roughly equal expression of ADAMTS4 and LRP1 in the amnion epithelium at the mRNA levels, and considerably higher expression of ADAMTS4 than LRP1 in the amnion fibroblasts; the protein data from 2G shows extremely high amounts of LRP1 in the amnion epithelium and roughly equal amounts of LRP1 and ADAMTS4 in the amnion fibroblasts. This seems like much more of a difference between mRNA and protein expression than I'm used to seeing, and likely unaccounted for by housekeeping genes. All samples are taken from TNL amnion, including the AE and AF in vitro cell culture. Are the data labels in 2F correct? It would make more sense if AE and AF data were swapped AND the ADAMTS4 and LRP1 data were also swapped. If all the data is accurate, please comment on the large discrepancy between the mRNA and protein data in the results or discussion.

Reply: We are sorry for confusion here. Please see our reply to question 3 from the editor for this issue.

Reviewer #2: The authors beautifully showed that ADAMTS4-mediated versican degradation amplifies the cytokine release and degrades collagen fibrils during labor in the amnion by the sophisticated experiments using primary human amnion cells and pregnant mouse model. The manuscript is well-written. However, there are some concerns in the experiments of Figure 5B and 8, so major revision is necessary.

Major Comments:

1. Line 191. Please explain the background and the roles of serum amyloid A1 (SAA1) in relation to the membrane rupture and labor because this molecule is not familiar to readers.

Reply: Thank you for reviewing our manuscript and all the constructive comments. Our previous studies have demonstrated that human fetal membranes are capable of synthesizing serum amyloid A1 (SAA1), an acute phase protein of inflammation which is thought to be synthesized primarily in the liver. We found that SAA1 expression was significantly increased in the human amnion at parturition, wherein SAA1 participated in parturition by inducing a number of inflammation mediators including IL-1 β , IL-6, IL-33 and PGE₂ (Li WJ et al, Sci Rep, 2017, 7: 693; Lei WJ et al, Mol Med 2023, 29: 88.). Moreover, SAA1 also participated in membrane rupture at parturition by inducing the degradation of collagen I via both autophagic and MMP pathways (Wang WS et al, Clin Sci (Lond), 2019, 133: 515-

530.). *Intraperitoneal injection of SAA1 induced preterm birth in the mouse (Gan XW et al, Front Immunol, 2020, 11: 1038.)*. Following your suggestion, the background information of SAA1 has been introduced briefly in the revised manuscript. Please see Page 8, Lines 190-193.

2. Lines 202-204. The experiment of Figure 5B and its interpretation is confusing. This Figure 5B presumably wants to show that endocytosis of ADAMTS4 does not occur if LRP is knocked down. First, the reason why recombinant ADAMTS4 was used is unclear (Figure 5B, 2nd and 4th rows). I think the authors should use cytokines (IL-1 β) instead to induce ADAMTS4. Immunofluorescence of ADAMTS4 in the cells treated with recombinant ADAMTS4 is nonsense. So I recommend using IL-1 β instead of recombinant ADAMTS4 in this knockdown experiment and immunofluorescence. Second, was EEA1 expression decreased by si-LRP1? This is because the expression change of EEA1 between NC (1st row) and si-LRP1 (3rd row) is unclear from this image. Please explain the interpretation of the Figure 5B experiment.

Reply: Thank you for raising this issue. You are right that Figure 5B is meant to illustrate the involvement of LRP1 in the endocytosis of ADAMTS4 in amnion fibroblasts. According to our data, IL-1 β could induce LRP1 shedding in amnion fibroblasts (Figure 4E), which would make the observation of ADAMTS4 endocytosis via LRP1 difficult. Instead, we used recombinant ADAMTS4 to illustrate the involvement of LRP1 in ADAMTS4 endocytosis in this LRP1 knockdown experiment, which has also been reported in many previous studies (Yamamoto Kazuhiro et al, FASEB J, 2013, 27: 511-2; Yamamoto Kazuhiro et al, J Biol Chem, 2014, 289: 6462-6474).

Early endosome antigen 1 (EEA1) is a marker for early endosomes in endocytosis. Decreased co-localization of ADAMTS4 with EEA1 with knockdown of LRP1 meant less endocytosis, indicating that internalization of ADAMTS4 was blocked. As a matter of fact, there was no change of EEA1 expression with si-LRP1 (3rd row in Figure 5B), which is in agreement with our Western blotting data (as shown below). Following your suggestion, we provided higher magnification images in Figure 5B to better illustrate the idea in the revised manuscript.

3. Lines 212-214. The authors suggest the reciprocal expression change of versican and verkiln by IL-1 β and SAA1 is mediated by ADAMTS4. If the authors want to prove the importance of ADAMTS4, an additional experiment might be necessary that amnion

fibroblasts are treated by cytokines under the existence of ADAMTS4 inhibitor and see the expression of versican and versikine.

Reply: This is an excellent point. Following your suggestion, we treated amnion fibroblasts with IL-1 β in the absence or presence of an ADAMTS4 inhibitor, and found that versican degradation induced by IL-1 β was attenuated by the ADAMTS4 antagonist. Please see the data in the revised Figure S4, which indicates the involvement of ADAMTS4 in the IL-1 β -induced versican degradation in amnion fibroblasts.

4. Lines 234-237. The results of the experiments of intra-amniotic injection of a lentiviral vector expressing Adamts4 (Fig. 8) should be carefully interpreted, although the experiment itself is sophisticated. This preterm birth was induced by forced expression of Adamts4, not by versican-versikine. Therefore, it is possible that preterm birth and concomitant cytokine increase was induced by other molecules induced by Adamts4 than versican-versikine. Proteases such as ADAMTS4 have potency to directly induce cytokines which might have caused preterm birth in this model. In other words, this experiment does not prove the direct versican-versikine effect to pregnant mice. Why did not the authors inject versikine itself into mice instead of ADAMTS4? This experiment alone cannot lead to the conclusion as “the observed effect of intra-amniotic administration of ADAMTS4 on parturition in the mouse is derived largely from its proteolytic effect on VCAN” (lines 275-278).

Reply: You are absolutely right. We do not have the data to confirm the direct involvement of VCAN cleavage in parturition at the current stage. Although our animal study showed that intra-amniotic injection of lentiviral vectors expressing Adamts4 induced preterm birth along with reciprocal changes in VCAN and versikine. At the current stage, we can only conclude that ADAMTS4 plays a crucial role in parturition, and ADAMTS4-mediated VCAN cleavage is possibly one of the underlying mechanisms. We have modified the title, abstract and conclusion to reflect this notion in the revised manuscript.

Minor comments:

1. Line 43. Abbreviations are typically defined the first time the term is used. It might be better to change “VCAN” here to “versican”.

Reply: We agree. All the abbreviation are now defined the first time when the term is used.

2. Line 53. The term “preterm premature rupture of the membranes” was recently changed to “preterm prelabor rupture of the membranes”. It might be better to change “premature” to “prelabor”.

Reply: Thank you for your suggestion. This term is now changed accordingly in the revised manuscript.

Reviewer #3:

1. Please specify the amnion cell type used (epithelial or mesenchymal) for the study in the abstract.

Reply: Thank you for your comments. We have now specified amnion fibroblasts for the study in the abstract.

2. The rationale for choosing Hyalactan, from a mixture of collagen and non-collagen-rich extracellular matrix (ECM) proteins, must be highlighted. This may include a breakdown of the % of hyalactan family members in the membrane matrix as they seem minimal. This is critical to support the hypothesis.

Reply: *This is an excellent point. We agree that the non-fibrous hyalactans only comprise a small portion of the extracellular matrix, which does not mean that they are not important. As a matter of fact, hyalactans, by forming a loose hydrated and malleable matrix with hyaluronan, control numerous normal and pathological processes in the extracellular matrix, including ECM remodeling, morphogenesis, cell signaling and immune response (Nandadasa Sumeda et al, Matrix Biol, 2014, 35: 34-41). Given the role of fibrous collagens but not the non-fibrous hyalactans in membrane rupture is very well recognized, we focused on the non-fibrous hyalactans in this study.*

3. Also, a description of how they may impact membrane matrix architecture is also important. Inflammation often seems to drive the production of matrix proteins as an innate mechanism to maintain homeostasis. Please include a short description on how this will impact the hypothesis.

Reply: *You are absolutely right. Inflammation could drive the production of matrix proteins as an innate mechanism to maintain homeostasis, such as tissue fibrosis (Jasso Guadalupe J et al, PLoS Biol, 2022, 20: e3001532.). However, it is very well recognized that inflammation can also disrupt membrane integrity resulting in membrane rupture at parturition, which is very well illustrated in infection-induced chorioamnionitis. Following your suggestion, this point of view is now fairly introduced in the introduction. Please see Page 4, Lines 103-104.*

4. Fig 1. Of the nine samples, maybe few (~3) samples show increased versikine and the others are somewhat similar to TNL. Specifically, one outlier maybe driving the stats. Please comment and may downplay the role of versikine.

Reply: *Thank you for raising this issue. After standardization with internal loading control (β -actin), 6 of 8 samples show increased versikine abundance in the TL group compared to the samples in the TNL group (as shown in the right column of Figure 1C). This may be caused by the heterogenous nature of human tissue samples. Although one outlier may drive the statistics, the P value still reaches 0.05 when this outlier is removed. However, we do not have the right reason to remove this outlier.*

5. Fig 1D: A higher magnification image would help to determine the staining pattern in the matrix region. Looks like some nonspecific staining in the matrix region in versikine.

Reply: *Please see our reply to question 2 from the editor for this issue. We performed immunohistochemical staining of versikine in human amnion tissue with further dilution of the antibody and a short incubation time. We believe that the staining is specific. The images are now replaced and higher magnification images are provided in the revised manuscript.*

6. Fig 2: Since fibroblast of the amnion are increased during TL, do authors consider the increase in ADAMTS4 is a remnant of an expected increased fibroblast? This can also be an argument that authors should do that the increased fibroblasts and subsequent increase in ADAMTS4 is a physiologic response required at term to facilitate degradation of VCAN. May discuss this.

Reply: *This is an excellent point. According to our single-cell sequencing data, ADAMTS4 abundance was increased in amnion fibroblasts at the single cell resolution in term labor, suggesting that the increase in ADAMTS4 in amnion tissue is more likely due to an increase in ADAMTS4 expression in amnion fibroblasts although the increase in the number of fibroblasts may also be a contributing factor in vivo.*

7. The rationale for choosing fibroblast for the studies needs explanation. Fig 1D shows a major shift in versikine in amnion epithelial layer and VCAN is scattered throughout the matrix (as expected) and not necessarily in the fibroblast. I would have chosen epithelial cells to study these factors or both cell types.

Reply: *This is an interesting and excellent point. It is very well recognized that amnion fibroblasts are the major cell type responsible for the production of matrix components. In this study, we also demonstrated that ADAMTS4 was produced primarily by amnion fibroblasts. In addition, VCAN, the target of ADAMTS4, was found abundantly distributed in the mesenchymal layer in which the fibroblasts are embedded but not in epithelial cells. In addition, amnion fibroblasts are important sources for inflammatory cytokines and PGE2 in parturition. All these reasons justified the use of amnion fibroblasts in this study. However, it remains an interesting issue to explore in the future why there appeared a shift of versikine, the cleavage product of VCAN, to epithelial cells in labor. Since the epithelial cells undergo senescence prior to parturition, it is tempting to postulate that the uptake of versikine may be associated with the senescence process of epithelial cells given that versikine is a bioactive damage-associated molecular pattern (DAMP) (Schmitt Michael, Blood, 2016, 128: 612-3).*

8. Transitioning amnion epithelial cells maybe a major and better source of versikine and that may give a good indication of how cellular transitions help to promote localized inflammation (as authors rightfully claimed) and promote ECM degradation.

Reply: *Although versikine can be expressed and secreted through stably transfected cells by plasmid construction and transfection (McCulloch Danie et al, Dev Cell, 2009, 17: 687-98), versikine is normally produced by VCAN proteolysis in the ECM (Nandadasa Sumeda et al,*

Matrix Biol, 2014, 35: 34-41; Timms Katherine et al, Glycobiology, 2020, 30: 365-373.), rather than originating from a particular type of cells. As we replied to your question 7, we assume that the shift of versikine to epithelial cells in labor may reflect the increased uptake of versikine from VCAN cleavage rather than increased versikine synthesis in epithelial cells.

9. Fig 4: Having the mRNA and proteins on the same scale does not make much sense. Please explain how normalization was done.

Reply: We are sorry for confusion here. Histograms of Figure 2A and B showed the fold change of ADAMTS4 mRNA or protein compared to their own control group in the previous version of manuscript. Now we split the mRNA and protein data into two graphs in order to make the data clearer and understandable in the revised manuscript.

10. Fig 4: The western blot panels don't show any changes with IL-1b and SAA, although there is a significant change. May explain how the normalization was done.

Reply: For normalization, the ratio of targeted proteins to corresponding internal loading controls for each group was compared to that of the control group. Here are all the original Western blotting images, which showed similar change in ADAMTS4 with IL-1β or SAA1 treatment. To better represent our data, we replaced the represent blots in the revised manuscript.

11. The data on endocytosis of ADAMTS4 is interesting

Reply: Thank you for the comment.

REVIEWERS' COMMENTS:

Reviewer #1 (Remarks to the Author):

The authors have addressed all of my concerns.

Reviewer #2 (Remarks to the Author):

The authors replied to all my questions and comments by performing additional experiments and adding explanations. The revised manuscript is now significantly improved and it is worth publishing. Thank you for giving me an opportunity to review this interesting paper.